



# 1 Towards the North Sea wind power revolution

Jens N. Sørensen[1], Gunner C. Larsen[2]
[1] DTU Wind Energy, Fluid Mechanics, 2800 Lyngby, Denmark
[2] DTU Wind Energy, Wind Turbine Loads and Control, 4000 Roskilde, Denmark
*Correspondence to*: Jens N. Sørensen (jnso@dtu.dk)
**Abstract.** The present work assesses the potential of a massive exploitation of offshore wind power in the North Sea by
combining a meteorological model with a cost model that includes a bathymetric analysis of the water depth of the North
Sea. The overall objective is to assess if the wind power in the North Sea can deliver the total consumption of electricity
in Europe and to what prize as compared to conventional onshore wind energy. The meteorological model is based on the
assumption that the exploited area is so large, that the wind field between the turbines is in equilibrium with the
atmospheric boundary layer. This makes it possible to use momentum analysis to determine the mutual influence between
the atmospheric boundary layer and the wind farm, with the wind farm represented by an average horizontal force
component corresponding to the thrust. The cost model includes expressions for the most essential wind farm cost
elements, such as costs of wind turbines, support structures, cables and electrical substations, as well as operation and
maintenance as function of rotor size, interspatial distance between the turbines, and water depth. The numbers used in
the cost model are based on previous experience from offshore wind farms, and is therefore somewhat conservative. The
analysis shows that the lowest energy cost is obtained for a configuration of large wind turbines erected with an interspatial
distance of about eight rotor diameters. A part of the analysis is devoted to assessing the relative costs of the various
elements of the cost model in order to determine the components with the largest potential for reducing the cost price. As
an overall finding, it is shown that the power demand of Europe, which is 0.4 TW or about 3500 TWh/year, can be fulfilled
by exploiting an area of 190.000 km$^2$, corresponding to about 1/3 of the North Sea, with 100.000 wind turbines of
generator size 13 MW on water depths up to 45m at a cost price of about 7.5 €cents/kWh.

## 24 1 Introduction

Although offshore wind energy has grown significantly over the past years, it only contributes with about 3% of the total
deployed wind energy. Measured by the investments and effort by the European wind energy industry to reduce the cost
of offshore wind power, it also is clear that offshore wind power will become a very important part of the future European
power production. As an illustration of this (see Fig. 1), 15 new offshore wind farms are at the moment under development
in Europe, contributing with an installed capacity of more than 4.000 MW, and in addition many offshore wind farms are
planned in the European seas (The European offshore wind industry, 2016). An important question is to what extent the
North Sea can be exploited with respect to a massive penetration of wind turbines, and what are the economic aspects of
doing this. As an overall objective, we here address the question if the North Sea can deliver the total consumption of
electricity in Europe and to what prize. To answer these questions it is required to determine the available wind resources
as well as the associated costs of erecting and operating wind turbines in the ocean. The first question regarding the
available wind resources is not trivial, as the presence of the turbines due to mutual wake effects alters the local wind
conditions. Hence, erecting wind turbines close to each other will reduce the wind speed and by this the efficiency of the
total power production. On the other hand, if the turbines are too far from each other, the full potential of the wind
resources in the North Sea will not be achieved. The most important parameter in this context is the mutual distance




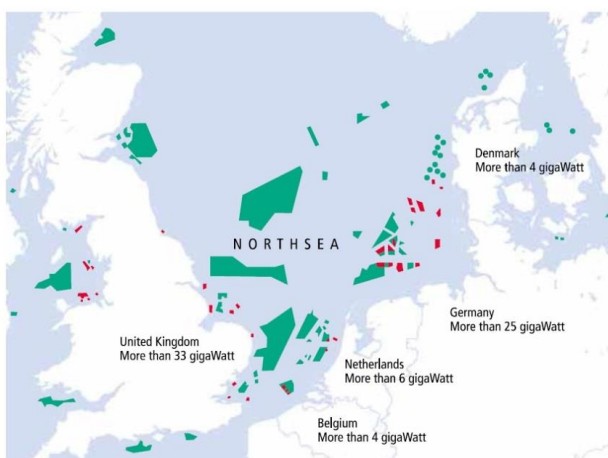

Figure 1: Planned and realized wind farms in the North Sea (Source: Offshore Wind, Clean Energy from the sea – Chris Westra, December 2014).

between the turbines, measured in rotor diameters, which is a reference length for wind farms. Today, in a typical wind farm, such as Rødsand or Horns Rev, the turbines are located 6-7 diameters from each other in order to diminish the wake effects. However, in a very large wind farm covering a main part of the North Sea this number may be different. Another important parameter is the size of the turbines, measured in installed generator power, or, alternatively, in rotor diameter. While the size of wind turbines erected onshore, due to visual impact, noise and other issues related to the lack of public acceptance, has stabilized on a maximum of about 3.5 MW, the size of wind turbines erected offshore is still increasing because of the influence of size on the reduction of cost of energy, which is much more pronounced offshore than onshore. Today, the biggest offshore wind turbines have a diameter of more than 160 m and an installed generator capacity of 8 MW. An important parameter in a cost analysis of offshore wind turbines is water depth, as the price of foundations and substructures heavily depends on water depth. Therefore, an economic analysis requires to be complemented with a bathymetric analysis. Other important economic parameters are costs of installation as well as operation and maintenance, both of which are substantially increased because of the harsh weather conditions appearing in the North Sea.

In the following, we address the various issues related to a massive penetration of wind power in the North Sea, including an assessment of the available wind power, the bathymetry of the North Sea, and an economic analysis. As wind farm design parameters we employ the interspatial distance between the turbines, measured in rotor diameters, and the turbine size, which here is varied in the range from about 3 MW to 20 MW. Furthermore, to simplify the analysis, issues and constraints, like fishery, sailing routes, political aspects, etc., are not taking into consideration. These aspects are certainly of importance, but outside the scope of the present analysis.

The paper is organized as follows. In chapter 2 we introduce the theory for the employed models, which is divided into a model for the power production and a model for the economic assessment of the installation. In section 3 results are shown and discussed, and in section 4 we conclude and outline the main findings.

## 2    Theory

The aim of this study is twofold – 1) to assess the wind *power area density* dependency on wind turbine size and spacing;





and 2) to determine the optimal wind turbine *size* and *interspacing* (i.e. wind farm topology) from an economic
perspective. The economic analysis is based on relatively simple models of foundation costs, cost of wind turbines, cost
of internal wind farm electrical infrastructure, and costs of operation and maintenance (O&M). Costs of lifetime fatigue
degradation of turbine components has, however, been neglected, but could, in a first order approximation, be considered
proportional to O&M costs. A more detailed approach is described by Rethoré et al. (2016), where cost of component
fatigue degradation is estimated using aeroelastic simulations of individual wind farm turbines exposed to unsteady wake
affected inflow conditions modeled using the Dynamic Make Meandering model (Larsen et al., 2008). In the following
subsections we describe and discuss the models used for wind resource estimation, for wind farm layout and for the cost
estimates on which the economic optimization will be based.

**2.1 Ressource estimation**
The model we employ to assess the wind power resource was originally developed by Templin (1974) and later developed
further by Frandsen and Madsen (2003) (see also Frandsen, 2005). The model is based on the assumption that the wind
farm is so large, that the wind field inside the wind farm is in equilibrium with the atmospheric boundary layer (ABL).
This makes it possible to use momentum analysis to determine the mutual influence between the atmospheric boundary
layer and the wind farm, with the wind farm represented by an average horizontal force component, corresponding to the
thrust, and the relative distance between the turbines as the main parameters. In the model it is assumed that the influence
of the wind turbines create two logarithmic boundary layers, which are connected at hub height by the shear forces exerted
by the turbines on the flow. The model results in the following simple equation to determine the mean velocity at hub
height inside the wind farm
$$U_h = \frac{G}{1 + \ln\left(\dfrac{G}{f \cdot h}\right)\dfrac{\sqrt{c_t + \left(\kappa / \ln\left(h / z_0\right)\right)^2}}{\kappa}} \ .$$
(1)

Here $G$ denotes the geostrophic wind speed, $h$ is the hub height of the wind turbines, with all turbines assumed to
be of equal size, and $f = 2\,\Omega \sin \varphi$ is the Coriolis parameter, in which $\Omega$ denotes the rotational speed of the earth, and $\varphi =$
55° (i.e. taken as the average latitude of the North Sea). The von Kármán constant is taken as $\kappa = 0.4$, and $z_o$ is the surface
roughness of the sea surface. The dimensionless parameter $c_t$ denotes the influence from the presence of the wind turbines
on the deceleration of the wind speed inside the wind farm. This parameter is given by the following expression
$$c_t = \frac{\pi C_T}{8S^2} \ ,$$
(2)

where $C_T$ is the thrust coefficient at which the wind turbine is operating, and $S = L/D$ denotes the dimensionless distance
between the turbines, measured in turbine diameters, $D$.
In the following some of the parameters in the model will be simplified in order not to complicate the study
unnecessarily. In general, the wind speed in the ABL depends on the vertical distance from the ground or sea surface,
following the logarithmic law for neutral stability conditions. The parameters that govern the deceleration of the wind
speed due to the presence of the turbines are, as can be seen from eq. (2), the density of the turbines, i.e. how close they
are located from each other, and the axial load, i.e. the thrust coefficient. For simplicity, it is here assumed that *the hub*
*height is equal to the rotor diameter*, and that the turbines operate close to the optimum, which here is taken as $C_T = 0.8$.
Furthermore, in the following the average undisturbed wind speed is taken as 9.7 m/s at 100 m height, corresponding to



a geostrophic wind speed of 12.2 m/s, and a roughness length at the sea surface $z_o = 0.001$m, numbers that are considered
realistic for the North Sea (Penna and Hahmann, 2017 and Hahmann, 2017). By using eq. (1), the decelerated wind speed
can be determined for different park turbine densities.
In general, the average distance between wind turbines in existing offshore wind farm corresponds to 6D – 8D. In
some wind farms, however, such as the Swedish Lillgrund wind farm, the distance may be as low as 3.3D. The denser
the turbines are located, the more the wind speed will be decelerated, which reduces the efficiency of the wind farm. On
the other hand, a large distance between the turbines means a less total exploitation of the wind resource within the wind
farm area.  In the following analysis the distance between the turbines is taken as one of the two main variable parameters
– the other being the turbine size.
**2.2  Average power production**
In order to determine the wind farm power production as well as to provide input to the applied cost model for wind farm
operation and maintenance expenses, we need to estimate the ambient mean wind speed statistics as well as the associated
wind farm mean wind speed statistics.

**2.3  Average production under ambient conditions**
Ambient wind speed statistics over the year (typically based on 10 minute or 30 minute averaging periods) are traditionally
quantified using a two-parameter Weibull distribution. The probability density function (pfd) of a Weibull distributed
random variable is
$$f\left(x;\lambda,k\right)=\begin{cases} \dfrac{k}{\lambda}\left(\dfrac{x}{\lambda}\right)^{k-1} e^{-\left(\frac{x}{\lambda}\right)^{k}} & ;x\geq 0 \\ 0 & ;x<0 \end{cases} \tag{3}$$

where $x$ is a realization of a stochastic variable $X$, $k > 0$ is the Weibull shape parameter, and $\lambda > 0$ is the Weibull scale
parameter.
The power production of a solitary wind turbine, $P(U)$, at a given mean wind speed $U$ may, below rated wind speed
$U_r$, be approximated by the following generic expression
$$P\left(U\right)=\alpha U^{3}+\beta , \tag{4}$$

which obviously allows for zero turbine production at cut-in wind speed $U_{in}$. Including this constraint in addition to the
rated (installed) generator power $P_r$ , with $U_r$ denoting the rated wind speed, the coefficients are determined as
$$\alpha = \frac{P_r}{U_r^3 - U_{in}^3} , \qquad \beta = -\frac{P_r U_{in}^3}{U_r^3 - U_{in}^3} , \tag{5}$$

The definition of the power coefficient gives the following relation between rated power, rotor diameter, $D$, and rated
wind speed,

$$P_r = \frac{1}{8}\rho\pi D^2 U_r^3 C_{P,rated} , \tag{6}$$


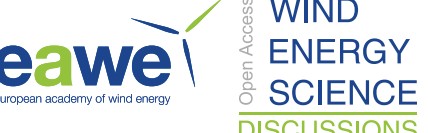



with $\rho$ being the air density and $C_{P,rated}$ the rated power coefficient, which here is taken as 0.5. We assume that the wind
turbine operates at its optimum condition at wind speeds lower than the rated wind speed and at a constant power yield at
wind speeds higher than the rated one. This is typical for a modern wind turbine, which is operated with variable tip
speed at low wind speeds below the rated one, and which is pitch-regulated at higher wind speeds. With these assumptions
the wind turbine power curve is given as

$$P(U) = \begin{cases} \alpha U^3 + \beta & ; \ U_{in} \le U < U_r \\ P_r & ; \ U_r \le U \le U_{out} \end{cases}. \tag{7}$$



where the wind turbine cut-out wind speed is denoted as $U_{out}$. In the analysis it is assumed that $U_{in} = 3$ m/s and $U_{out} = $
25 m/s. The average production of the wind turbine, $P_y$, may be formulated as a convolution of the wind turbine production
characteristics with the mean wind speed probability density function expressed in eq. (3). Thus

$$\begin{aligned} P_y &= \int_{U_{in}}^{U_{out}} P(U) f(U;\lambda,k) dU \\ &= \alpha \int_{U_{in}}^{U_r} U^3 f(U;\lambda,k) dU + \beta \int_{U_{in}}^{U_r} f(U;\lambda,k) dU + P_r \int_{U_r}^{U_{out}} f(U;\lambda,k) dU \end{aligned}. \tag{8}$$


Reformulating the Weibull distribution, eq. (3), as

$$f(U;\lambda,k) = \begin{cases} \dfrac{-d}{d(U)} e^{-\left(\frac{U}{\lambda}\right)^k} & ; x \ge 0 \\ 0 & ; x < 0 \end{cases}, \tag{9}$$


eq. (8) simplifies to
$$P_y = \alpha \int_{U_{in}}^{U_r} U^3 f(U;\lambda,k) dU + \beta \left( e^{-\left(\frac{U_{in}}{\lambda}\right)^k} - e^{-\left(\frac{U_r}{\lambda}\right)^k} \right) + P_r \left( e^{-\left(\frac{U_r}{\lambda}\right)^k} - e^{-\left(\frac{U_{out}}{\lambda}\right)^k} \right). \tag{10}$$

The remaining integral in eq. (10) is solved using the variable transformation, $t = \left(\dfrac{U}{\lambda}\right)^k$, whereby we obtain

$$\int_{U_{in}}^{U_r} U^3 f(U;\lambda,k) dU = \lambda^3 \int_{(U_{in}/\lambda)^k}^{(U_r/\lambda)^k} t^{3/k} e^{-t} dt = \lambda^3 \left[ \Gamma\left(\frac{3+k}{k}, \left(\frac{U_{in}}{\lambda}\right)^k\right) - \Gamma\left(\frac{3+k}{k}, \left(\frac{U_r}{\lambda}\right)^k\right) \right], \tag{11}$$


where $\Gamma(*,*)$ is the Incomplete Gamma function (cf. Abramowitz and Stegun, 1970, p.260). Finally, introducing (11) in
(10) we obtain the following closed form expression for the average wind turbine production






$$P_y = \alpha\lambda^3\left[\Gamma\left(\frac{3+k}{k},\left(\frac{U_{in}}{\lambda}\right)^k\right) - \Gamma\left(\frac{3+k}{k},\left(\frac{U_r}{\lambda}\right)^k\right)\right] + \beta\left(e^{-\left(\frac{U_{in}}{\lambda}\right)^k} - e^{-\left(\frac{U_r}{\lambda}\right)^k}\right) + P_r\left(e^{-\left(\frac{U_r}{\lambda}\right)^k} - e^{-\left(\frac{U_{out}}{\lambda}\right)^k}\right) \ . \ (12)$$


The Weibull parameters depend in general on altitude as well as on the stability conditions of the ABL. For the present
North Sea study we simplify matters by assuming neutral ABL stability condition "in average", and under this assumption
we conjecture that the Weibull shape parameter is *independent* of altitude. The mean of the Weibull distribution (i.e. the
yearly mean wind speed), $\bar{U}_y$, may be expressed as
$$\bar{U}_y = \lambda\Gamma\left(1+1/k\right), \tag{13}$$

where $\Gamma$ (*) is the Gamma function. As seen, $\bar{U}_y$ scales directly with the Weibull scale parameter for a fixed shape
parameter. As scale parameters we employ $\lambda = 11$ m/s and $k = 2.2$, corresponding to an average wind speed of 9.7 m/s, at
a 100 m altitude. The numbers are taken as averaged values from measurements and simulations of selected locations in
the North Sea (see Pena and Hahmann, 2017).
Discharging non-neutral atmospheric boundary layer stability conditions, a logarithmic shear profile may be assumed,
meaning that the *relative* increase in mean wind speed, $f_{\Delta U}$, for an increase in altitude from a reference height $z_r$ to
height $z$ is given by
$$f_{\Delta U} = \bar{U}/\bar{U}_{ref} = Ln\left(z/z_0\right)/Ln\left(z_{ref}/z_0\right), \tag{14}$$

with $z_0$ being the roughness length and $\bar{U}_{ref}$ being the mean wind speed at the reference height.
The wind turbine capacity factor, $f_C$, expresses the ratio of the actual yearly output to its potential output, if it were possible
to operate at full nameplate capacity continuously over the year. For the solitary turbine it is accordingly defined as
$$f_C = P_y/P_r \ , \tag{15}$$

with $P_y$ obtained from eq. (12).
Assuming that the Weibull shape parameter is independent of altitude, the formulas for turbine average production (eq.
(12)) and capacity factor (eq. (15)) apply for *all* altitudes, if the Weibull scale parameter, $\lambda$, associated with a reference
height, is replaced with $f_{\Delta U}\lambda$ (cf. eq. (14)). In the above, the roughness length has implicitly been assumed constant,
which strictly speaking is true only for an on-shore site. For offshore conditions the surface roughness depends on the
wind speed, which complicates matters somewhat. However, this is disregarded in the present study.
**2.2.2 Average production under wind farm conditions**
The wind speed statistics inside a wind farm is different from the wind speed statistics of the ambient undisturbed flow
discussed in the previous subsection. This is due to the wind speed reduction caused by the wind turbines, which, for a
very large wind farm, may be estimated according to eq. (1). In this subsection, we will derive the distribution of the mean
wind speed at hub height inside an "infinite" wind farm and in turn estimate the average power production of turbines





operating inside the "infinite" wind farm. In analogy with the previous subsection, the estimate will be based on an
assumed Weibull distributed ambient mean wind speed at relevant hub heights, meaning that the Weibull scale parameter,
$\lambda$, may be adjusted by the factor defined in eq. (15) in case the hub height in question differs from the reference hub
height.
To proceed, we note that the mean wind speeds at hub height respectively inside and outside the wind farm are
described by two interrelated stochastic variables. We will consider the mean wind speed inside the wind farm as resulting
from a transformation of the ambient undisturbed mean wind speed according to the receipt described in Sec. 2.1. The
mean wind speed at hub height, $U_H$, inside the "infinite" wind farm is given by (cf. eq. (1)),

$$U_H = \frac{G}{1 + ln\left(\dfrac{G}{fh_H}\right)\dfrac{\sqrt{c_t + \left[\kappa / ln\left(h_H / z_0\right)\right]^2}}{\kappa}} \ . \tag{16}$$


For $c_t = 0$ we obtain the ambient wind speed at hub height as

$$U_{H,0} = \frac{G}{1 + ln\left(\dfrac{G}{fh_H}\right)\dfrac{1}{ln\left(h_H / z_0\right)}} \ . \tag{17}$$


We introduce the following short hand notation

$$\gamma = ln\left(\frac{G}{fh_H}\right), \qquad \delta = ln\left(\frac{h_H}{z_0}\right), \tag{18}$$

whereby

$$U_H\left[1 + \gamma\frac{\sqrt{c_t + \left(\kappa / \delta\right)^2}}{\kappa}\right] = U_{H,0}\left[1 + \frac{\gamma}{\delta}\right], \tag{19}$$

or
$$U_H = U_{H,0}\frac{1 + \dfrac{\gamma}{\delta}}{1 + \gamma\dfrac{\sqrt{c_t + \left(\kappa / \delta\right)^2}}{\kappa}} \ . \tag{20}$$

The thrust coefficient $C_T$ is approximated as

$$C_T = \begin{cases} C_{T,rated} \ ; & U_{in} \leq U_H < U_r \\ C_{T,rated}\left(U_r / U_H\right)^{3/2} \ ; & U_r \leq U_H \leq U_{out} \end{cases}, \tag{21}$$



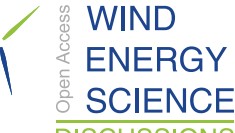
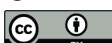

where $C_{T,rated}$ is the rated thrust coefficient, which in the following is taken as 0.8, and $U_r$ is the rated wind speed.
Introducing eq. (2) into eq. (20) one obtains

$$\frac{U_H}{U_{H,0}} = \begin{cases} \varepsilon_1 = \dfrac{1+\dfrac{\gamma}{\delta}}{1+\dfrac{\gamma}{\kappa}\sqrt{\dfrac{\pi C_{T,rated}}{8S^2}+\left(\kappa/\delta\right)^2}} \; ; & U_{in} \leq U_H < U_r \\[3em] \varepsilon_2 = \dfrac{1+\dfrac{\gamma}{\delta}}{1+\dfrac{\gamma}{\kappa}\sqrt{\dfrac{\pi C_{T,rated}}{8S^2}\left(U_r/U_H\right)^{3/2}+\left(\kappa/\delta\right)^2}} ; & U_r \leq U_H \leq U_{out} \end{cases} \qquad (22)$$




As seen from eq. (22), $\varepsilon_1$ is a constant whereas $\varepsilon_2 = \varepsilon_2(U_H)$ depends on the actual velocity at hub height.
To determine the probability density function for the wind farm, we exploit the following relationship between the original
Weibull distribution, $f_{H,0}$, and the altered distribution, $f_H$, due to the wake effects from the wind turbines in the farm,

$$f_H\left(U_H\right)dU_H = f_{H,0}\left(U_{H,0}\right)dU_{H,0}. \qquad\qquad (23)$$


The probability density function of $U_H$ in the below rated regime can now be formulated in closed form by combining eq.
(22) and eq. (23),

$$f_H(U_H) = f_{H,0}(U_{H,0})\frac{dU_{H,0}}{dU_H} = \frac{f_{H,0}\left(U_H/\varepsilon_1\right)}{\varepsilon_1}; \qquad U_{in} \leq U_H < U_r \quad . \qquad (24)$$

It is intuitively clear that, with the wind speed transformation expressed in (22) for the below rated regime, an infinitesimal
probability around $U_{H,0}$ for the ambient conditions, equals an infinitesimal probability around $U_H$ for the infinite wind
farm conditions, which is exactly what is expressed in eq. (24). As in section 2.2.1, we assume the *ambient mean wind*
speeds to be Weibull distributed (cf. eq. (3)), whereby we finally obtain the following mean wind speed probability density
function for the below rated *wind farm wind climate*,

$$f_H\left(U_H\right) = f_{H,0}\left(U_H;\varepsilon_1\lambda,k\right) \; ; \qquad U_{in} \leq U_H < U_r , \qquad (25)$$

which is a Weibull distributed mean wind speeds with scale parameter $\varepsilon_1\lambda > 0$.
We now turn to the *above rated wind farm regime*. Assuming again that the mean wind speed in the ambient domain is
Weibull distributed, the expected yearly wind turbine production for the above rated wind farm wind speed regime may





be formulated as

$$P_r \int_{U_r}^{U_{out}} f_H\left(U_H\right) dU_H = P_r \int_{U_r/\varepsilon_2(U_r)}^{U_{out}/\varepsilon_2(U_{out})} f_{H,0}\left(U_{H,0};\lambda,k\right) dU_{H,0}. \qquad (26)$$

or, using eq. (10)

$$P_r \int_{U_r}^{U_{H,o}} f_H\left(U_H\right) dU_H = P_r\left( e^{-\left(\frac{U_r}{\varepsilon_2(U_r)\lambda}\right)^k} - e^{-\left(\frac{U_{out}}{\varepsilon_2(U_{out})\lambda}\right)^k} \right). \qquad (27)$$

We are now ready to compute the yearly output of a wind farm turbine, which then in turn is used to determine the wind
farm capacity factor defined in eq. (15). Employing eq. (27), and otherwise taking a similar approach as the one leading
to eq. (12) for a solitary turbine, the yearly power output is determined as

$$P_{WF,y} = \int_{U_{in}}^{U_{out}} P\left(U_H\right) f_H\left(U_H;\varepsilon\lambda,k\right) dU_H$$

$$= \alpha \int_{U_{in}}^{U_r} U_H^3 f_H\left(U_H;\varepsilon_1\lambda,k\right) dU_H + \beta \int_{U_{in}}^{U_r} f_H\left(U_H;\varepsilon_1\lambda,k\right) dU_H + P_r\left( e^{-\left(\frac{U_r}{\varepsilon_2(U_r)\lambda}\right)^k} - e^{-\left(\frac{U_{out}}{\varepsilon_2(U_{out})\lambda}\right)^k} \right). \qquad (28)$$

The first two terms in eq. (28) can be determined analytically, in analogy with the derivation leading to eq. (12), and we
thus finally obtain the following closed form expression for the average annual power output of a wind farm turbine,

$$P_{WF,y} = \alpha\left(\varepsilon_1\lambda\right)^3\left[ \Gamma\left(\frac{3+k}{k},\left(\frac{U_{in}}{\varepsilon_1\lambda}\right)^k\right) - \Gamma\left(\frac{3+k}{k},\left(\frac{U_r}{\varepsilon_1\lambda}\right)^k\right) \right] + \beta\left( e^{-\left(\frac{U_{in}}{\varepsilon_1\lambda}\right)^k} - e^{-\left(\frac{U_r}{\varepsilon_1\lambda}\right)^k} \right)$$

$$+ \quad P_r\left( e^{-\left(\frac{U_r}{\varepsilon_2(U_r)\lambda}\right)^k} - e^{-\left(\frac{U_{out}}{\varepsilon_2(U_{out})\lambda}\right)^k} \right). \qquad (29)$$

Essentially, it is only allowed to exploit eqs. (23) and (24) if it can be proved that there exists a one-to-one transformation
between $f_H$ and $f_{H,0}$. A way to prove this is to demonstrate that $U_H = U_H(U_{H,0})$ is a monotonic function. For the
*below rated wind speed* case this is easily shown as $\varepsilon_1$ in eq. (22) is a constant. For the *above rated wind speed* case a
formal proof is given in App. A.
**2.4 Wind farm layout characteristics**
The specific wind farm topology assumed for the present study is the simplest possible; i.e. a quadratic grid with the wind
turbines uniformly interspaced in two perpendicular horizontal directions. Hence, assuming a total number of wind
turbines, $N_T$, located at a distance, $L$, from each other, the side length of the quadratic wind farm grid is given
as $L\left(\sqrt{N_T} - 1\right)$. With this assumption, the required area, $A$, relates to the number of turbines as

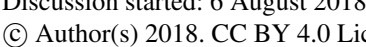
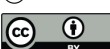


$$ A = \left[ L \left( \sqrt{N_T} - 1 \right) \right]^2 = S^2 D^2 \left( \sqrt{N_T} - 1 \right)^2 , \tag{30} $$

Where $S$ is the wind turbine interspacing in rotor diameters, $D$. For a given area the number of turbines is from eq. (30) determined as

$$ N_T = \left[ \frac{\sqrt{A}}{S_L D} + 1 \right]^2 . \tag{31} $$

To determine the installed capacity we will need a relationship between the turbine rated power, $P_r$, and the rotor diameter. This is obtained by assessing eq. (6) at rated wind speed,

$$ P_r = K \cdot D^2 , \tag{32} $$

where

$$ K = \frac{\pi}{8} \rho U_r^3 C_{P,rated} . \tag{33} $$

Combining eqs. (30) and (32), we get the following expression for the power area density (i.e. the installed capacity per area unit)

$$ \frac{N_T P_r}{A} = \frac{N_T K}{S^2 \left( \sqrt{N_T} - 1 \right)^2} . \tag{34} $$

**2.5 Bathymetry of the North Sea**

The North Sea is nearly 1000 km long and 600 km wide, with a total area of around 570.000 km$^2$. Most of the North Sea is on the European Continental shelf and has an average depth of about 90 m. In the southern part the water is very shallow, with average water depths of 25 to 35 m, increasing to depths up to between 100 and 200 m north of the Shetland Islands. In the south, the depth is at most 50 m, and a large part of it is the sand bank Dogger Bank, which has water depth of about 25 m. Therefore, the southern part of the North Sea is ideal for erecting wind turbines.

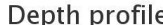
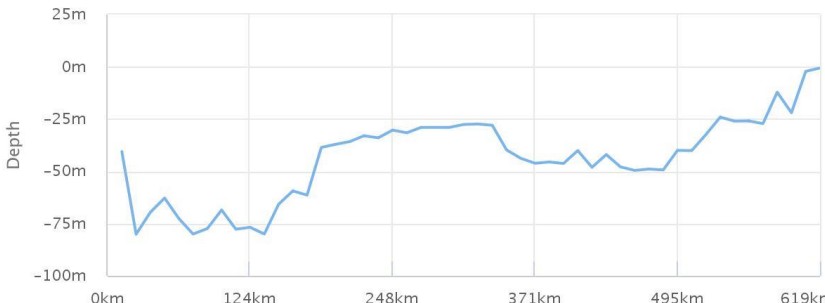

**Figure 2: Depth profile of a line spanning from New Castle (UK) to Hanstholm (DK).**

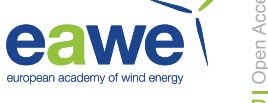 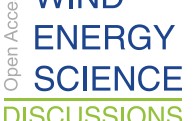

The bathymetric properties of the North Sea can be determined by inspection of the European Marine Observation
and Data Network (EMODnet, 2017). An example of this is shown in Fig. 2, where the water depth has been extracted
along a line going from the east coast of U.K (New Castle) to the west coast of Denmark (Hanstholm). As seen on the
plot a large part is covered by a shallow plane, which is the Dogger Bank. By systematically extracting data from this
website it has been made possible to generate the full bathymetric properties of the North Sea (Nielsen, 2015). This is
shown in Figs. 3 and 4, which depict the distribution of area (Fig. 3) and the accumulated area (Fig. 4) as function of
water depth. From these figures it is seen that about 250.000 km$^2$ of the sea has water depths less than 60 m, which makes
this part ideal for erecting wind turbines on monopoles or jacket substructures. A full mapping of the water depth is
important for the subsequent economic analysis, as the cost of wind turbine substructures depends heavily on water depth.

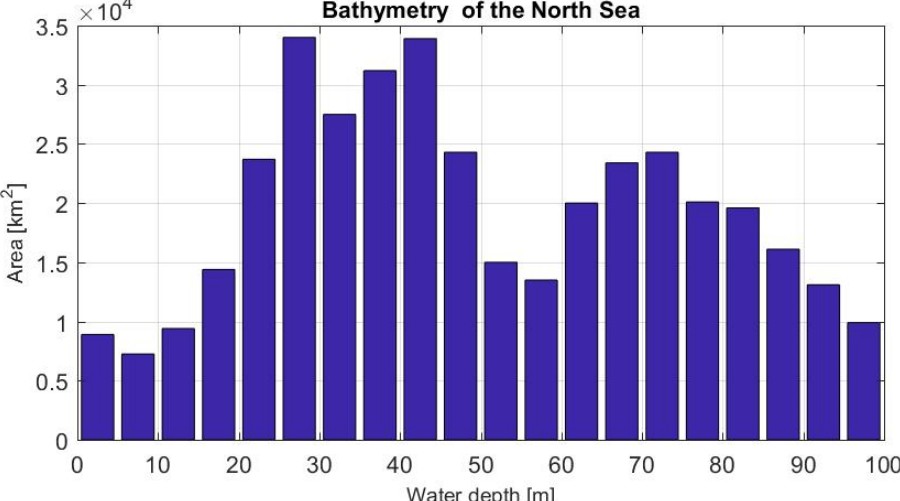


**Figure 3: Bar diagram showing areas with specific water depths.**

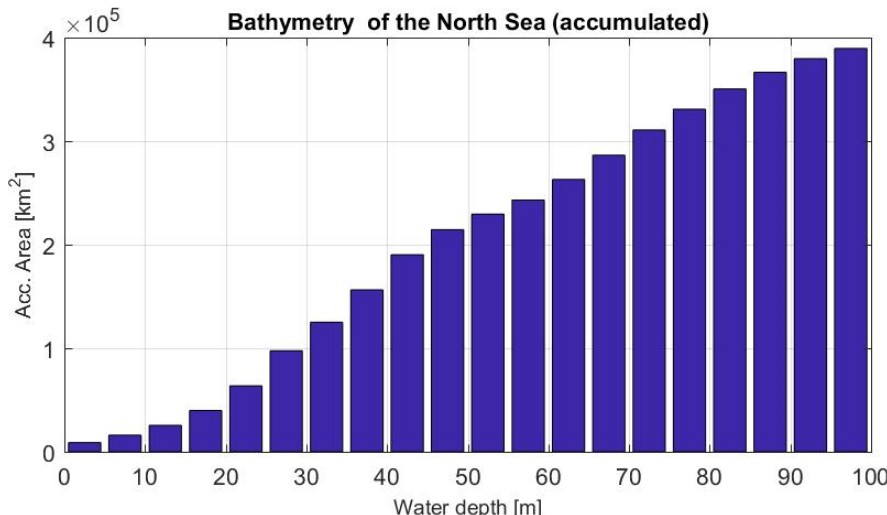


**Figure 4: Accumulated area as function of increasing water depth.**





**2.5. Cost models**
Cost models are needed for any economic optimization aiming at finding the optimal balance between wind turbine
production, operational costs and financial costs. Given the broad and generic character of the present study, relatively
simple models have been used. These, as well as the assumptions on which they are based, are described in the following.

**2.5.1 Cost of wind turbine**
The cost of a wind turbine in M€, $C_{WT}$, may according to Lundberg (2003) be taken as $C_{WT} = -0.15 + 0.92 P_R$, where
$P_R$ is the installed generator power in MW. However, this pricing refers to the year 2003, where the report was compiled.
The inflationary development in (Danish) consumer prices in general from 2003 and up to the year 2015 is 23% (Retail
prices index, 2015). In this study we will assume wind turbine prices to follow the inflation in general consumer prices
during this period, and we will further add 2% to approximately include the wind turbine price development up to today
(i.e. 2017). With these assumptions we finally arrive at the following expression for wind turbine prices in M€

$$C_{WT} = 1.25\left(-0.15 + 0.92 P_R\right). \tag{35}$$


**2.5.2 Cost of support structure**
Cost and type of wind turbine support structures depend primarily on wind turbine size and water depth. A monopole
foundation is considered advantageous for shallow water regimes, which in the present context means water depths up to
about 35m. For water depths beyond 35m jacket foundations are convenient and consequently assumed.
The cost of a *monopile* support structure in M€, $C_{FM}$, may in a first order approximation be simplified as (Buhl
and Natarajan, 2015)
$$C_{FM} = \frac{P_R\left(H^2 + 100H + 1500\right)}{7500}, \tag{36}$$


where $P_R$ denotes the wind turbine rated power in MW, and $H$ is the water depth in meters.
Cost of a *jacket* support structure in M€, $C_{FJ}$, may in a first order approximation be simplified as (Buhl and
Natarajan, 2015)
$$C_{FJ} = \frac{P_R\left(4.5H^2 - 35H + 2500\right)}{7500}. \tag{37}$$

**2.5.3 Cost of wind farm electrical grid**
Assuming the internal electrical grid predominantly (i.e. except for one connecting line along the alternative direction)
laid out along one of the directions in the quadratic grid, the aggregated length of the grid cables, $L_C$, is given by
$$L_C = SD\left(\sqrt{N_T} + 1\right)\left(\sqrt{N_T} - 1\right) = SD\left(N_T - 1\right). \tag{38}$$



The wind farm grid financial costs pr. running meter, including cable cost and costs of installation, for an offshore site is
taken as $C_C$ =675€ (Rethoré et al., 2014 and Larsen et al., 2011). Consequently, the total aggregated grid costs, $C_G$, are
given as
$$C_G = L_C C_C .\tag{39}$$

**2.5.4 Cost of operation and maintenance**
Cost of operation and maintenance (O&M), $C_{O\&M}$, depends on turbine size as well as on wind turbine spacing, in
the sense that a smaller spacing, and thereby higher loadings, increases the costs and, for larger turbines, these costs are
reduced per installed MW. It is reasonable to assume that the *relative* wind turbine size effect (e.g. the relative reduction
in O&M for one 6MW wind turbine compared to two 3 MW WT's) for wind turbines subjected to identical load conditions
is independent of the particular load level, and we will consequently assume that the size and load dependencies can be
factorized as

$$C_{O\&M}\left(P_R,S\right) = f_{WT}\left(P_R|P_{R,Ref}\right)\cdot C_{WT_{Ref}}\cdot f_C \cdot f_S\left(S\right),\tag{40}$$


where $f_{WT}\left(P_R|P_{R,Ref}\right)$ is the wind turbine size factor, $C_{WT_{Ref}}$ is the yearly cost of O&M for a reference turbine with rated
power, $P_{R,Ref}$ , operating under ideal conditions with a wind turbine capacity factor equal to one, $f_C$ is the wind turbine
capacity factor for an imaginary solitary wind turbine at the site of interest, and $f_S\left(S\right)$ is a load factor accounting for
the impact of the wind farm load level, and thus of the wind turbine spacing, on the O&M costs. The load factor depends
on the load condition for the particular wind farm turbine, and it is expressed in terms of wind farm topology (i.e. spacing)
as

$$f_S\left(S\right) = \frac{P_{S,y}}{P_{WF,y}} = \left(\frac{P_{S,y}}{P_r}\right)/\left(\frac{P_{WF,y}}{P_r}\right) = \frac{f_C}{f_{WF}},\tag{41}$$


where $P_{S,y}$ is the average annual power yield of a solitary turbine at the site of interest, $P_{WF,y}$ is the average annual power
yield of a wind farm turbine and $f_{WF} = P_{WF,y} / P_r$ is the wind farm capacity factor. As seen, the load factor increases for
decreasing wind farm capacity factor (and vice versa) reflecting increased wake impact and thus in turn increased loading.
Inspired by Berger (2013), where a 14% reduction of annual OPEX cost per MW is stated by shifting from 3 MW
to 6 MW turbines, we will assume that this *relative* reduction can be *linearly* extrapolated to other WT sizes within the a
size regime spanned by half and double the size of the reference wind turbine, respectively. Outside this size regime it
seems reasonable to assume an exponential behavior, where 14% reduction of OPEX is gained for a doubling of wind
turbine size, and a corresponding increase of OPEX results if the wind turbine size is halved. Thus, for an increase in
wind turbine size

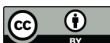

$$f_{WT}\left(P_R|P_{R,Ref}\right) = \begin{cases} 1 - \dfrac{0.14\left(P_R - P_{R,Ref}\right)}{P_{R,Ref}} & \text{for } P_{R,Ref} \le P_R \le 2\,P_{R,Ref} \\ 0.86^{0.5\,P_R/P_{R,Ref}} & \text{for } 2\,P_{R,Ref} < P_R \end{cases}. \qquad (42)$$


For a decrease in WT size the analog expression is

$$f_{WT}\left(P_R|P_{R,Ref}\right) = \begin{cases} 1 - \dfrac{0.325\left(P_R - P_{R,Ref}\right)}{P_{R,Ref}} & \text{for } 0.5\,P_{R,Ref} \le P_R \le P_{R,Ref} \\ 0.86^{-0.5\,P_{R,Ref}/P_R} & \text{for } P_R < 0.5\,P_{R,Ref} \end{cases}. \qquad (43)$$


Note, that the difference in factors in the linear expressions relates to the reference turbine being the smallest respectively
the largest turbine in these expressions.
The reference turbine is for the present study taken as a 10MW turbine, for which the O&M costs per year may be
specified as $C_{WT,Ref}$ = 106 €/kW (Chaviaropoulos and Natarajan, 2014).
Because O&M costs are running costs, contrary to the financial costs described in sections 2.5.1 – 2.5.3, which
refer to the time of the wind farm installation, we need assumptions on the development of O&M costs over time in
comparison with the inflation. We will here assume that the development of O&M costs over time follows the inflation
in general. This makes the rate of inflation the natural choice for the discounting rate, and with this choice we conveniently
avoid computation of net present values by letting all prices referring to the time of wind farm installation (Larsen, 2009).
**2.5.5 Levelized cost of energy**
Other costs than those described in the previous sections – e.g. cost of transformer station(s) and establishment of
a main cable to the coast – are presumed to depend only on the rated production of the wind farm and thus for the present
study independent of the wind farm *layout* (i.e. wind turbine spacing) and the choice of *turbine size*. Consequently, this
cost can in principle be omitted for the present layout cost optimization considerations. However, such costs will of cause
affect the levelized cost of energy (LCoE) estimate, and to arrive at reasonable realistic LCoE estimates we will, in line
with Mahulja (2015), assume that cost of WT's, internal WF grid and foundations accounts for 75% of the total investment
costs, which is based on experiences from the Danish Horns Rev and Nysted offshore wind farms. The remaining 25% is
mainly due to electrical infrastructures, such as onshore cables and substations. The estimated LCoE expressed in terms
of a kW price is consequently given by

$$LCoE = 1.33 \frac{N_T\left[C_{WT} + N_Y P_E C_{O\&M}/1.33 + \gamma C_{FM} + (1-\gamma)C_{FJ}\right] + C_G}{N_Y P_E}, \qquad (44)$$


where $\gamma$ is the fraction of wind turbines erected on monopole foundations, $(1-\gamma)$ is the fraction of wind turbines erected on
jacket foundations, $N_Y$ is the life time of the wind farm in years, and $P_E$ is the yearly consumption of electricity in kW.
For the present study we will assume a wind farm life time of 20 years; i.e. $N_Y$ = 20.




## 3    Results

As mentioned previously, the results include an investigation of the dependence of wind turbine size and interspacing on power density, as well as an analysis to determine the optimal wind turbine *size* and *interspacing* (i.e. wind farm topology) from an economic perspective. The economic model is formulated using a simple design space spanned by only two discrete optimization parameters, namely the mutual distance between the turbines, $S$, and the turbine size, $D$, which here is limited to take the values 100m, 150m, 200m and 250m. From eq. (6), assuming a rated wind speed $U_r = 11$ m/s, the turbine sizes are determined to correspond to an installed power of 3.2 MW, 7.3 MW, 13 MW and 20 MW, respectively.

### 3.1    Power density and area requirements

As a first part of the study we here analyze the power density of the wind resources in the North Sea and assess the area required to cover the power demand of Europe as per 2016. By solving the system of equations outlined in sections 2.1-2.3, the power density, i.e. the power production per unit area sea surface, may be obtained as a function of wind turbine spacing and rotor diameter. The outcome of this is shown in Fig. 5, which depicts the power density as a function of rotor spacing, $S$, spanning the range from 4 diameters to 11 diameters, and for the above mentioned four different rotor diameters. In this range it is seen that the power density decreases monotonically from about 4.5 W/m$^2$ at $S = 4$ to about 1 W/m$^2$ at $S = 11$. It should be noted that the power density attains a maximum at a rotor spacing of about $1.5D – 2D$, which, depending of rotor size, goes from 4 W/m$^2$ for $D = 100$m to 7.5 W/m$^2$ for $D = 200$m. For a 'standard' value of $S = 7$ and $D = 150$m, we get a power intensity of about 2 W/m$^2$. For a comparison, in a similar study by Frandsen et al. (2009), the power density was found to vary in the range form 1.9 W/m$^2$ to 4 W/m$^2$, depending on rotor size and spacing.

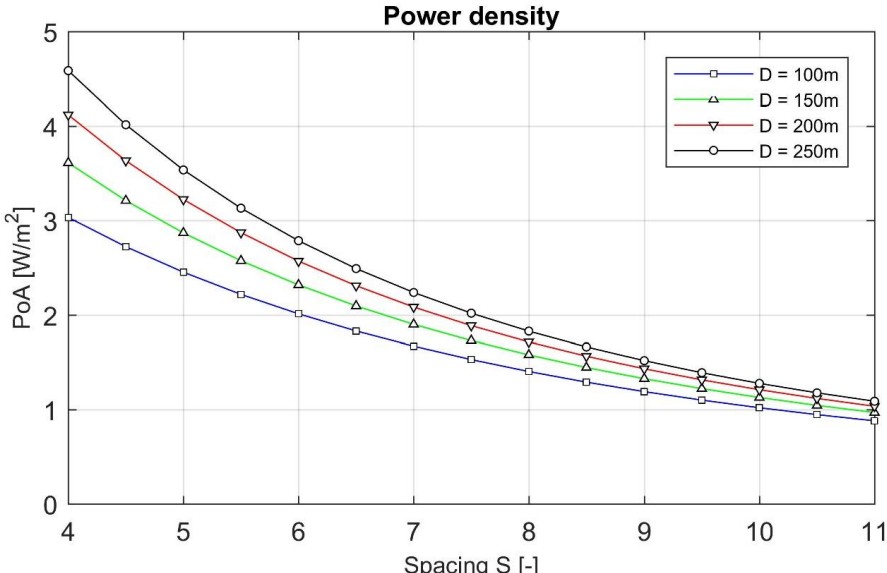

**Figure 5: Power density as function of spacing and rotor diameter.**

For existing wind farms, such as the Danish Nysted or Horns Rev wind farms, the power intensity is measured to range from 2.7 W/m$^2$ to 4 W/m$^2$ (Frandsen et al., 2009 and Volker, 2015). The corresponding capacity factor is in Fig. 6 seen to vary from about 0.15 to 0.4, again depending on turbine distance and diameter.





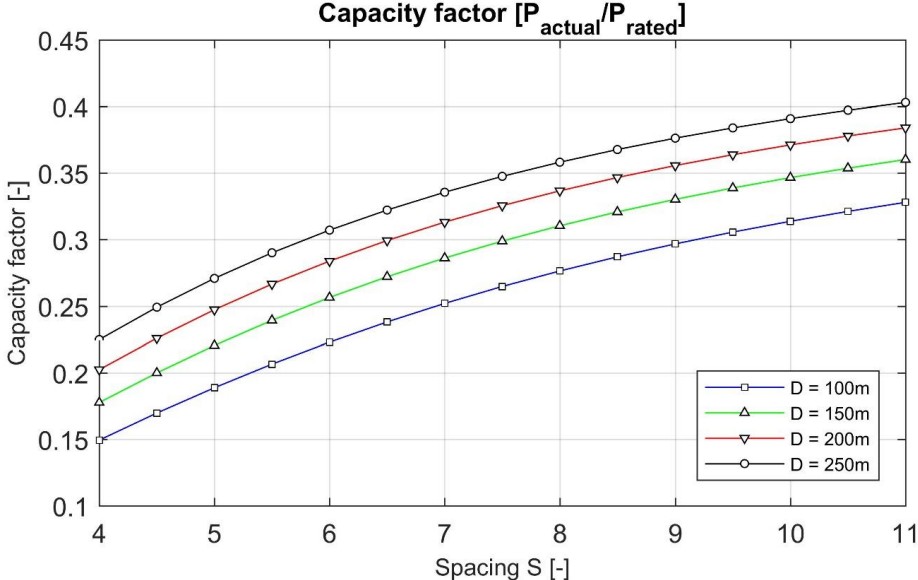


**Figure 6: Capacity factor as function of spacing and rotor diameter.**



The energy production in various parts of the North Sea is obtained by combining the bathymetry with the actual
annual energy production per area unit for a given combination of rotor size and interspacing. As an example, assuming
a rotor diameter D = 200m and a spacing S = 7, we get an energy production on different water depths as shown in Fig.
7. Essentially Fig. 7 is obtained by multiplying the values in Fig. 3 by the annual energy production per square kilometer,
as the energy production does not depend on the water depth.

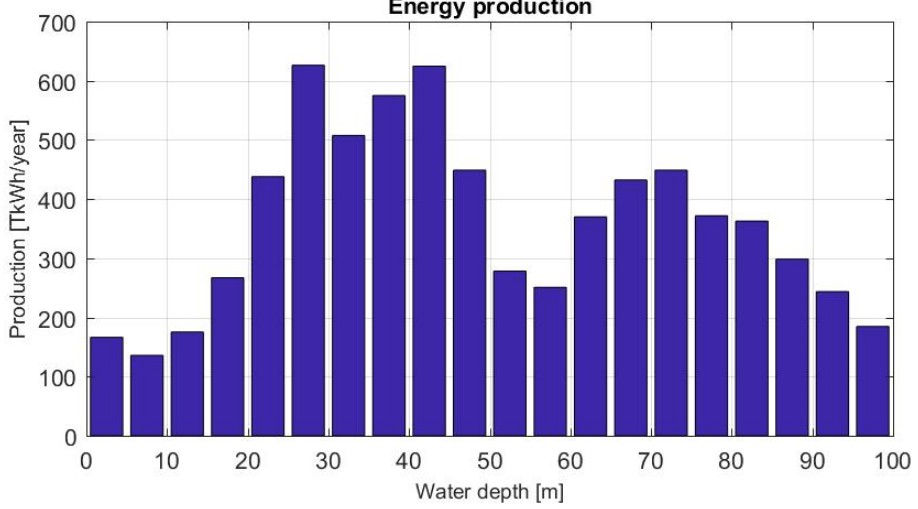


**Figure 7: Energy production as function of water depth for D=200m and S=7.**

The accumulated energy production on water depths is shown in Fig.8, which essentially is identical to Fig. 4,
except for a scaling of the ordinate. From the two figures it is seen, that most energy production in fact can be obtained

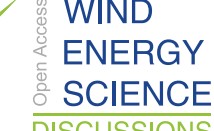

at relatively shallow waters depths. Hence, about half of the available wind energy of the North Sea may be harvested at
water depths below 45m.

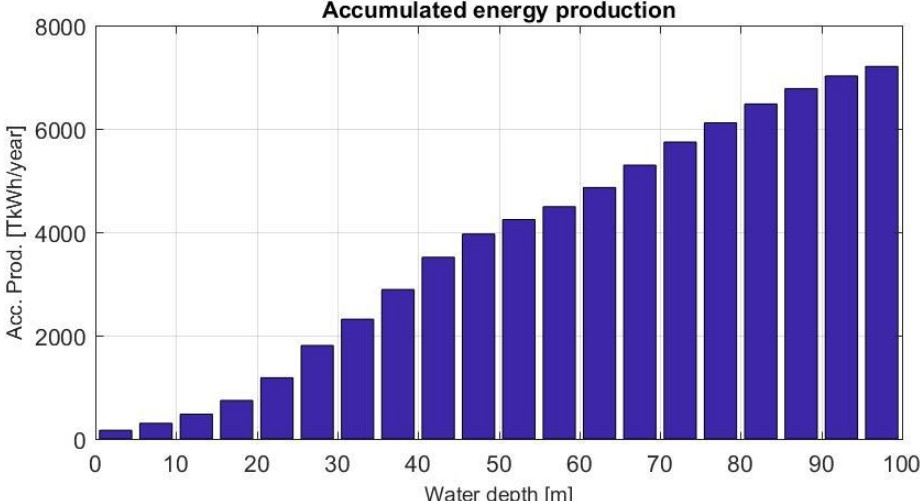


**Figure 8: Accumulated energy production as function of water depth for D=200m and S=7.**

Referring to the year 2016, the power demand for Europe is about 0.4 TW, corresponding to a production of about

3500 TWh/year (Eurostat Statistics Explained, 2016 and Electricity in Europe, 2013). Fig. 9 shows the area required to
provide the power demand for Europe as a function of wind turbine spacing and rotor diameter. For the chosen parameter
values, the required area is seen to be in the range from about 100.000 km$^2$ to about 450.000 km$^2$. For a foreseeable
'standard' configuration of $S = 7D$ and $D = 200$m, the required area is about 190.000 km$^2$. This corresponds approximately

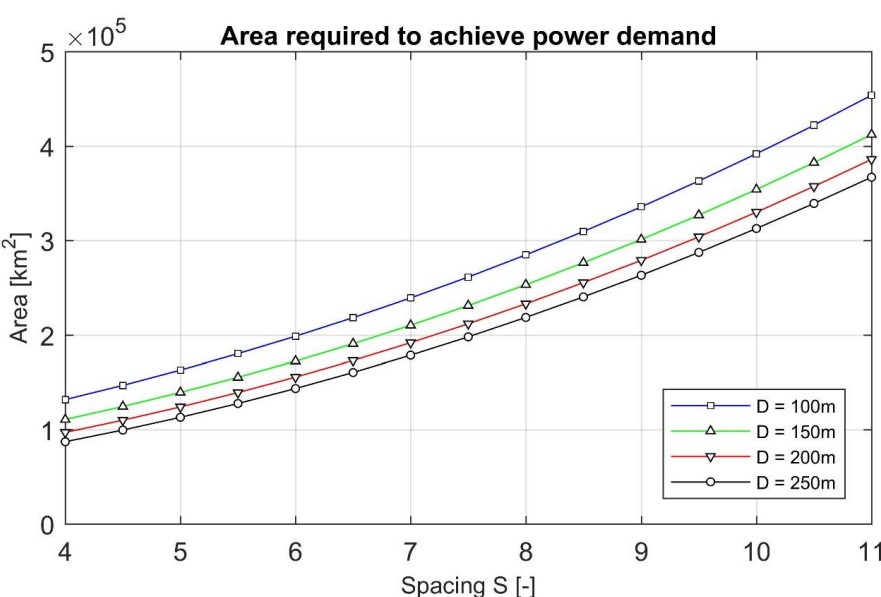


**Figure 9: Area required to produce Europe's power demand as function of spacing and rotor diameter.**



to 1/3 of the area of the North Sea and, as seen from Fig. 8, this target can be achieved by exploiting water depths less
than 45m. The required installed power and number of turbines are depicted in Figs. 10 and 11, respectively. For the
'standard' configuration it is required to install about 100.000 13 MW wind turbines, corresponding to an installed power
capacity of about 1.25 TW.

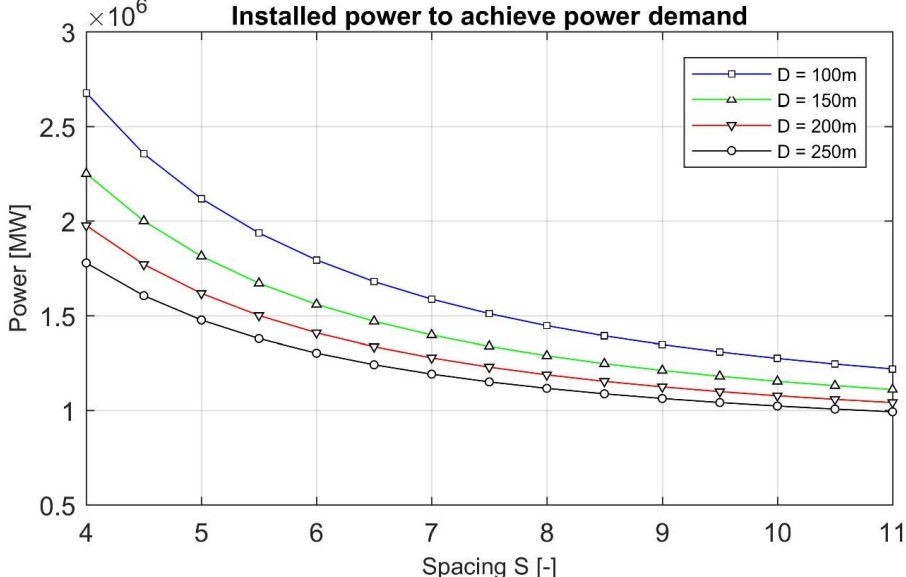


**Figure 10: Installed power required to produce Europe's power demand as function of spacing and rotor diameter.**

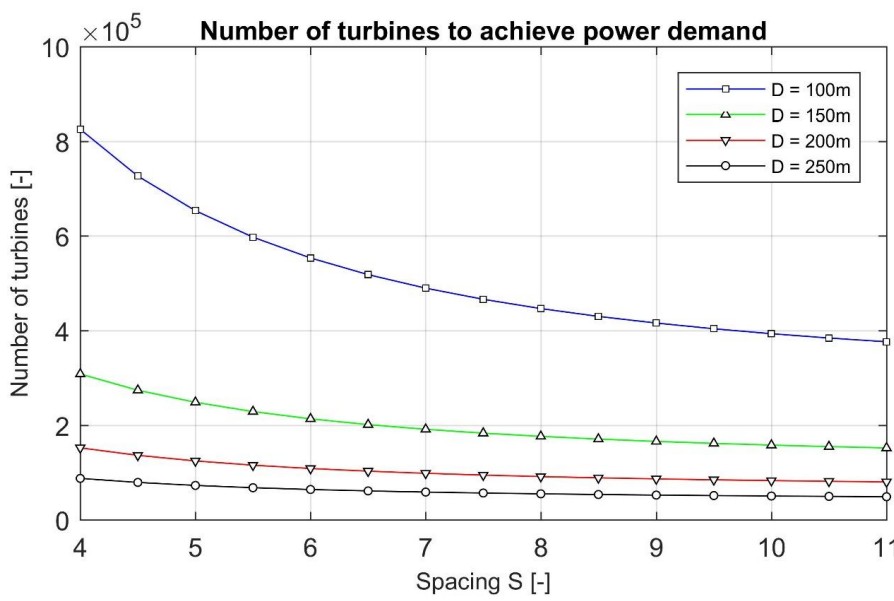


**Figure 11: Number of turbines required to produce Europe's power demand as function of spacing and rotor diameter.**



### 3.2 Economic analysis

Employing the various expressions of the cost model introduced in section 2.5, we here present and discuss the economic
aspects of a potential massive exploitation of wind power in the North Sea. As the foundation costs increase with water
depth, we will first exploit all available shallow sea bed area, and subsequently include successively deeper water regimes.

#### 3.2.1 Influence of water depth on cost of energy

By combining the bathymetry of the North Sea with the cost model it is possible to determine the relative cost of energy
as a function of water depth. In order to limit the number of variables we first assume a fixed rotor diameter D = 200m,
and then compute the LCoE for different wind turbine interspacing as a function of water depth. The result is shown in
Figs. 12, from where it is seen that the LCoE increases monotonously as a function of water depth, illustrating the added
expenses of the substructures at deeper waters. From Fig. 12 also seen that the LCoE reduces when placing the turbines

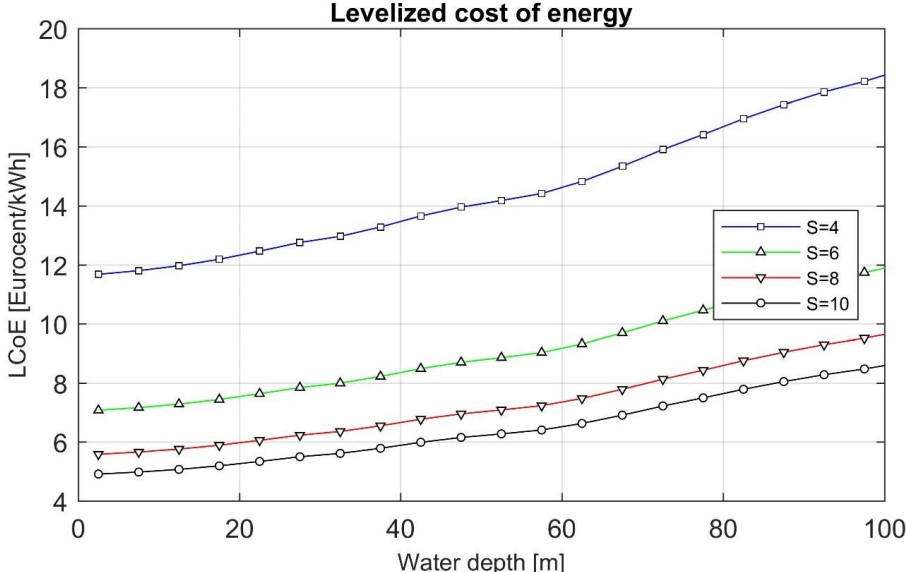

**Figure 12: Levelized cost of energy (LCoE) as function of increasing water depth for a 200m diameter rotor.**

further apart from each other, i.e. at increasing $S$-values. The reason for this is partly that the wind resources increase, as
wake effects becomes less pronounced at higher $S$-values, and partly that the O&M expenses decreases when erecting the
turbines further away from each other, also due to less pronounced wake effects. On the other hand the cable costs increase
when increasing $S$. However, this is less pronounced as compared to the decreasing cost effect of the wake effects. Fixing
the interspacing at $S = 8$ and varying the rotor size (Fig. 13), it is seen that the lowest cost of energy is obtained for the
biggest rotor size. This can partly be explained by increased wind resources, as the tower height increases for increasing
rotor diameters (it is implicitly assumed that the tower height equals the rotor diameter). From the figures, the LCoE is
seen to vary from about 5 €cents/kWh for large rotors located near the coast to nearly 13 €cents/kWh for smaller rotors
penetrating all water depths up to about 100m. As determined in section 3.1 it is required to exploit locations at all water
depth up to about 45m to comply with the electrical power demand of Europe. In this case the LCoE is found to be in the
range from 6 €cents/kWh to 9.5 €cents/kWh, depending on rotor size and the interspacing between the wind turbines.





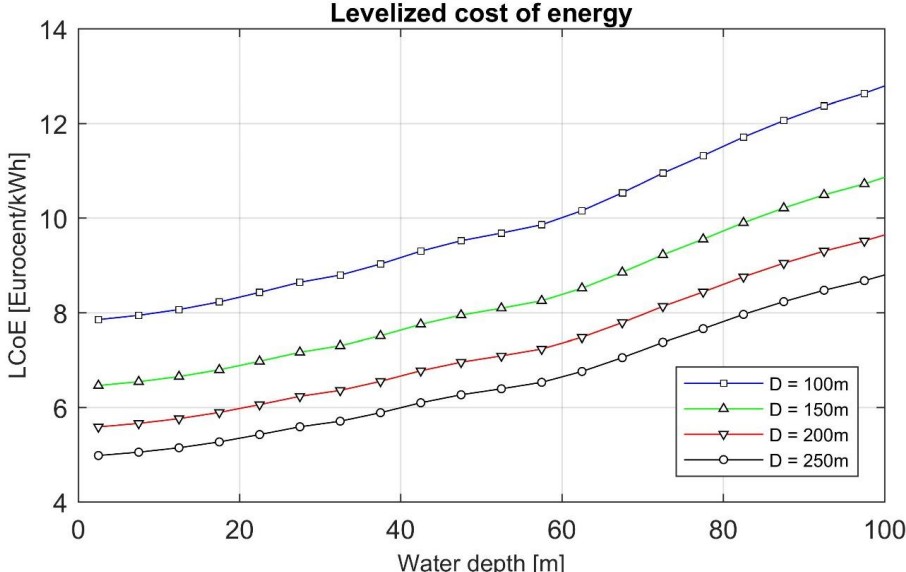

**Figure 13: Levelized cost of energy (LCoE) as function of increasing water depth for S = 8.**

It is interesting to put the computed cost estimates into perspective by looking at actual prices for existing wind farms.
For an existing wind farm such as Rødsand II, which has been in operation since 2010, the cost price is about 8
€cents/kWh. This wind farm, which covers an area of 35 km$^2$ located on shallow waters, consists of 90 2.3 MW wind
turbines of diameter 93 m (International Renewable Energy Agency IRENA working paper, 2012). This gives an average
distance between the turbines of about 7.5 diameters and a cost price of 62.9 øre/kWh (according to https://www.power-
technology.com/projects/rodsand). This cost price agrees very well with the curves shown in Fig. 13, where a wind farm
consisting of 100 m diameter wind turbines located at water depths up to 10 m produces wind power to an LCoE which
is exactly equal to 8 €cents/kWh. As seen in Fig.12, this price reduces with more than 30% just by increasing the rotor
diameter to 200 m.

**3.2.2 Cost of energy for covering the electricity need of Europe**
To determine the optimal combination of interspacing and rotor diameter for the required electrical power demand of
Europe, we compute the LCoE as function of wind turbine interspacing and rotor size for a fixed electrical energy
production of E = 3500 TWh/year. Here we have two counteracting phenomena. On one hand, LCoE decreases at
increasing interspacing between the turbines. On the other hand, increasing distances between the turbines demands more
space, and thus, in turn, more expensive grid installation costs are required, as well as the need to exploit the wind power
at locations on larger water depths, which then tends to increase the LCoE. It is therefore expected that there will be a
specific value of *S*, where the cost of energy attains a minimum. This is illustrated in Fig. 14, which depicts the LCoE as
a function of wind turbine interspacing and rotor diameter to comply with Europe's total electricity demand. It is here
seen that the lowest LCoE is obtained at an interspatial distance of about *S* = 8-9. It is also seen, that the lowest cost of
energy is obtained when increasing the rotor size. However, as mentioned above, this may partly be explained by the
increased wind resources at higher hub heights, as the tower height is assumed to be equal to the rotor diameter. From the

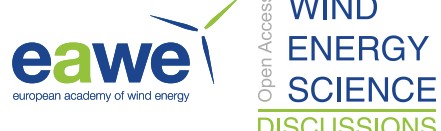

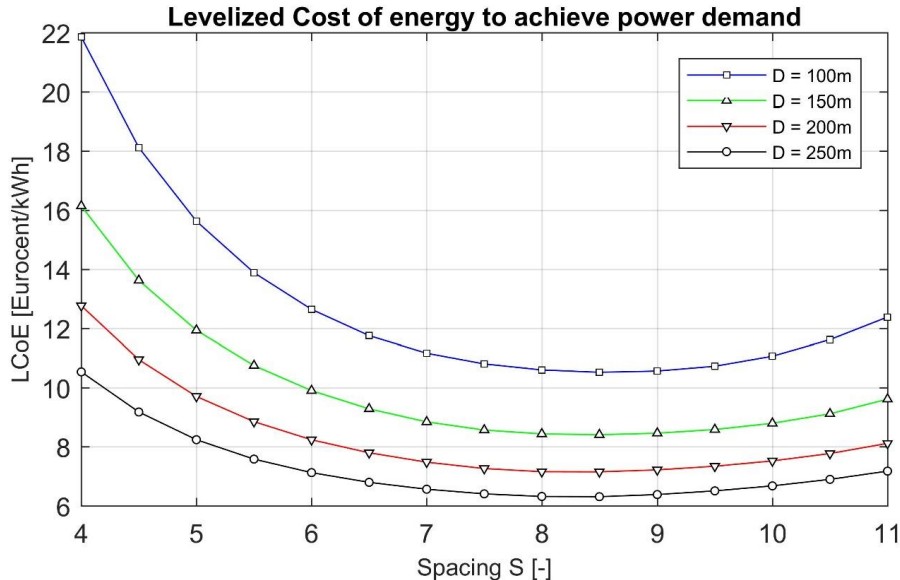


**Figure 14: Levelized cost of energy (LCoE) as function of wind turbine interspacing and rotor diameter to comply with Europe's electrical energy demand of E = 3500 TWh/year.**


figure it is seen that exploiting wind turbines of diameter D=250m, corresponding to an installed generator power of 20
MW, located with an interspacing of 8 diameters, results in an estimated cost price of about 6 €cents/kWh.

**3.2.3 Assessment of relative costs**
The relative cost of the various elements involved in offshore wind energy can be assessed from the cost models
introduced in section 2.5. In Fig. 15 we depict the relative costs on turbine, cables including substations, O&M, and

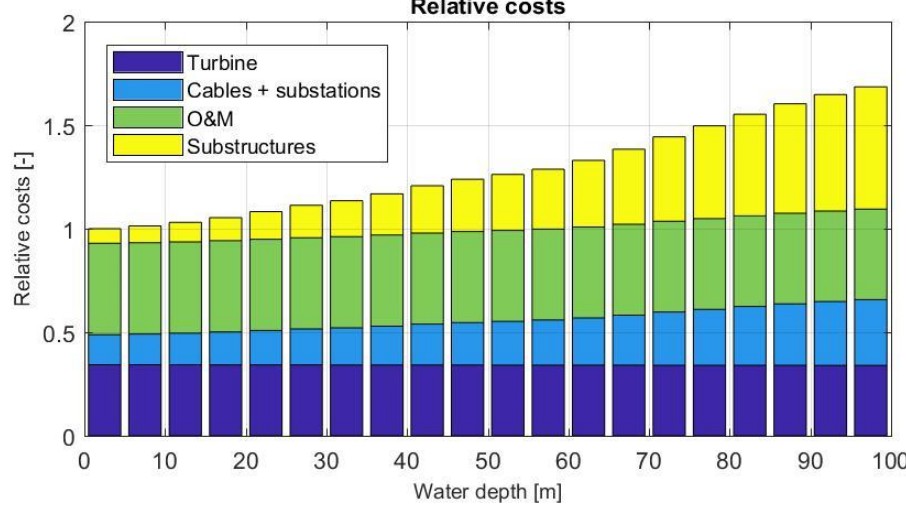


**Figure 15: Relative cost of wind turbine components as function of water depth for a farm configuration consisting of wind turbines of rotor diameter D = 200m and interspatial distance S = 7.**




support structures as a function on water depth for a farm configuration with rotor diameter D = 200m and interspatial
distance $S = 7$. The numbers are made dimensionless by the total cost of a turbine placed on the shallowest water. Hence,
the size of the bar at a given water depth refers to turbines placed on a reference water depth of h = 2.5m. It is here seen
that the total costs increases with about 20% when exploiting water depths up to 50m and with about 70% for water depths
up to 100m. It is here assumed that the support structures are limited to monopoles and jackets, following the cost model
described in section 2.5.2. As the interspatial distance between the turbines is fixed, the only cost that changes at different
water depths is the cost of the substructure and, to a lesser extent, the electrical substations. From the figure it is seen that
the relative cost of the substructures increases from about 5% of the total costs at h = 2.5m to about 20% at h = 50m.

Another way of assessing the relative costs is to fix the area and the rotor diameter and then determine the influence

of the interspatial distance between the turbines on the costs of the various items. In this case it is only the operation and
maintenance costs that change. This is shown in Fig. 16, which depicts the relative costs for a fixed rotor diameter D =
200m and a total exploited area A = 190.000km², again corresponding to water depths up to 45m. Note that the bars in

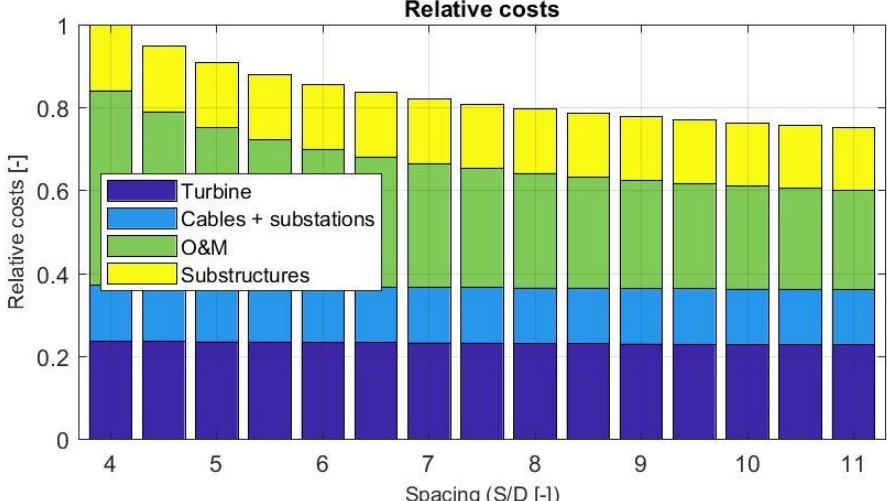


**Figure 16: Relative cost of wind turbine components as function of wind turbine interspacing**
**for a fixed rotor diameter D = 200m and area A = 190.000 km².**

Fig. 16 are made dimensionless with the total costs of the configuration with the smallest investigated interspatial distance
($S = 4$). It is here seen that the total costs decreases monotonously when increasing the interspatial distance from a
reference unit value at $S = 4$ to about 0.75 at $S = 11$. If we, as an example, take the relative cost prices at a configuration
with an interspatial wind turbine distance $S = 8$, which was the value with the lowest LCoE, we get that the cost of the
wind turbine amounts to 23%, the electrical substations including cables to 13%, the substructures to 17%, and the O&M
to 46% of the total costs. Hence, it is clear that the largest potential for reducing the cost price is to focus on reducing the
operation and maintenance costs.
**3.2.4 Cost considerations at a fixed area**
To assess the possibility of exploiting wind power at relatively shallow waters, we here fix the exploited area up to water
depths of 45m, corresponding to an area of the North Sea area of 190.000km², and compute the levelized cost of energy
as a function of wind turbine interspacing for various turbine sizes. The result is displayed in Fig.17, which shows that




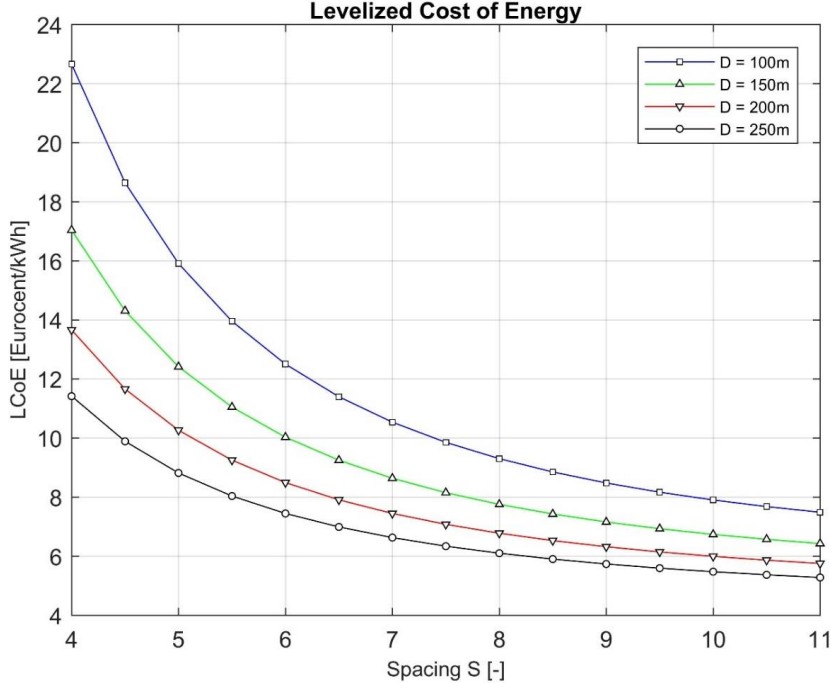


**Figure 17: Levelized cost of energy as function of wind turbine interspacing for various turbine sizes at a fixed area A = 190.000 km².**

the LCoE decreases monotonously when increasing the wind turbine interspacing. Assuming e.g. a rotor size D = 200m

the LCoE decreases from 14 €cents/kWh at $S = 4$ to 6 €cents/kWh at $S = 10$. Unfortunately, the total power yield also

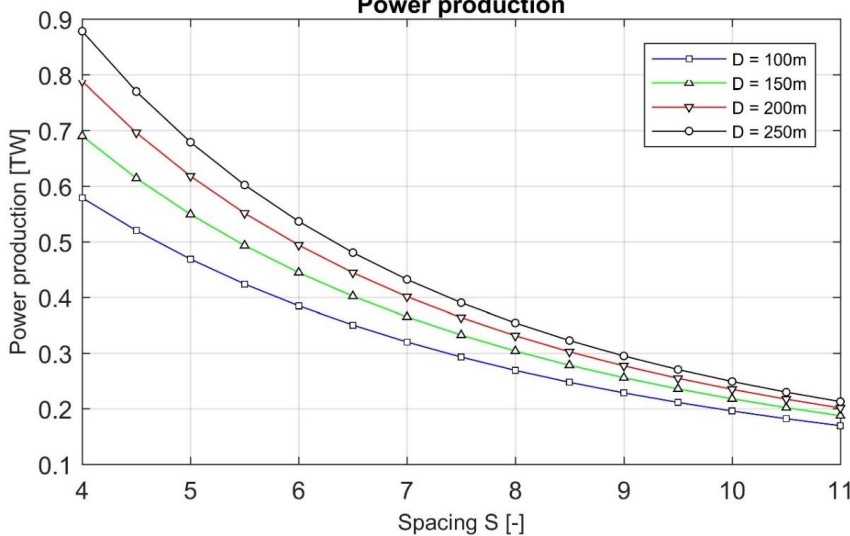


**Figure 18: Power production as function of wind turbine interspacing for various turbine sizes at a fixed area A = 190.000 km².**






decreases when increasing the distance between the turbines. This is shown in Fig. 18, which depicts the power production
as function of wind turbine interspacing for various turbine sizes at a fixed area A = 190.000km². Here it is seen that the
power yield for the same rotor size of D = 200m decreases from 0.8 TW at $S = 4$ at to 0.2 TW at $S = 11$. Combining the
two figures, one may determine the LCoE to achieve a specific power demand. This is shown in Fig. 19, which displays
the relative cost of energy as function of power demand for various turbine sizes, still assuming a fixed area A =
190.000km². It is seen that it is indeed possible to increase the power production to two times the present electrical power
demand of Europe and still only exploit an area of 190.000 km², corresponding to less than 1/3 of the area of the North
Sea. The price to pay, however, is that the levelized cost of energy increases from about 7.5 €cents/kWh to 14 €cents/kWh
for a configuration consisting of 200m diameter wind turbines with an interspacing S = 4. If North Sea instead only
provides a smaller part of the electricity demand for Europe, it is seen that the LCoE decreases correspondingly. As an
example, if the North Sea only is exploited to provide 50% of the European electricity demand, it is seen that the LCoE
may decrease to about 5.5 €cents/kWh for a 200 m rotor.

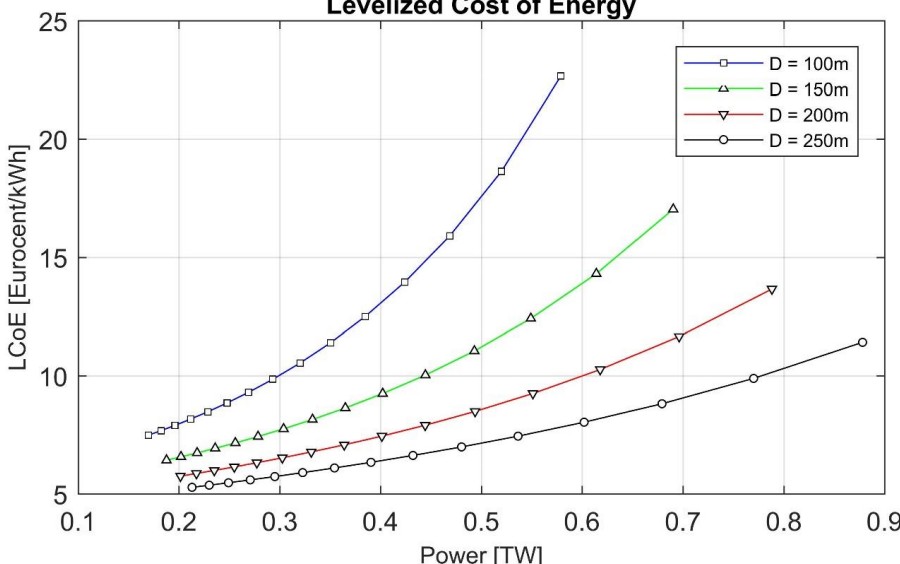


**Figure 19: Levelized cost of energy as function of power demand for various turbine sizes**
**at a fixed area A = 190.000 km².**


**4    Conclusions**
The present study focused on determining the potential of a massive exploitation of wind power in the North Sea. The
study combines a simple meteorological model for large wind turbine clusters (Templin, 1974 and Frandsen and Madsen,
2003) with an economic analysis including the bathymetry of the North Sea. The analysis comprises both an assessment
of the wind power potential in the North Sea and an estimate of the economics aspects associated with a large scale
exploitation of wind power in the North Sea. The main parameters of the model are wind turbine size, interspatial distance
between the turbines, and the area distribution on water depth. The analysis shows that the lowest cost of energy,
independent of the size of the turbines, is obtained at an interspatial distance of about eight rotor diameters between the
turbines. An important conclusion is that Europe's electrical power demand can be fulfilled by exploiting a surface area
of 190.00km² with wind turbines with a rotor diameter size of 200 m and with an interspatial distance of 8 diameters,





corresponding to 1.6 km. This corresponds approximately to 1/3 of the area of the North Sea and can be achieved by
exploiting water depths less than 45m. The required installed power corresponds to about 100.000 13 MW wind turbines
with a total installed power capacity of about 0.95 TW. Based on the presented cost model, the levelized cost of energy
then amounts to about 7.5 €cents/kWh. Replacing the 13 MW (D=200m) turbines with 20 MW turbines (D=250m),
reduces the cost price to 6 €cents/kWh.

Another part of the study concerned the relative cost of the various items involved in offshore wind energy. Here

it was found the operation and maintenance main contribute with up to 50% of the total expenses. Hence, the largest
potential for reducing the cost price is to focus on reducing the operation and maintenance costs.

Finally, it was found that it is possible to increase the power production to two times the present electrical power

demand of Europe and still only exploiting an area of 190.000 km$^2$, corresponding to less than 1/3 of the area of the North
Sea. The price to pay, however, is that the levelized cost of energy increases from about 6 €cents/kWh to 14 €cents/kWh
for a configuration consisting of 200m diameter wind turbines with an interspacing of four diameters.
**Acknowledgments**
The work has been carried out with the support of the Danish Council for Strategic Research for the project Center for
Computational Wind Turbine Aerodynamics and Atmospheric Turbulence (grant 2104-09-067216/DSF) (COMWIND:
http://www.comwind.org). The authors like to thank Bjarke Fuglsang Nielsen for providing bathymetric data for the North
Sea and Andrea Hahmann for providing the WRF-based North Sea Weibull parameters.

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



**Appendix A**
In this appendix the gradient of the wind farm mean wind speed, $U_H$ with respect to the ambient mean wind speed,
$U_{H,0}$ is proven to be positive in the above rated wind speed regime. From eq. (22) we have

$$U_H = U_{H,0} \frac{1 + \dfrac{\gamma}{\delta}}{1 + \dfrac{\gamma}{\kappa}\sqrt{\dfrac{\pi C_{T,rated}}{8S^2}\left(U_r / U_H\right)^{3/2} + \left(\kappa/\delta\right)^2}} \ , \qquad (A.1)$$

or
$$U_{H,0} = U_H \left(1 + \frac{\gamma}{\delta}\right)^{-1} \left(1 + \frac{\gamma}{\kappa}\sqrt{\frac{\pi C_{T,rated}}{8S^2}\left(U_r / U_H\right)^{3/2} + \left(\kappa/\delta\right)^2}\right). \qquad (A.2)$$

The gradient is thus expressed as

$$\begin{aligned} \frac{dU_{H,0}}{dU_H} \ = \ & \left(1 + \frac{\gamma}{\delta}\right)^{-1}\left(1 + \frac{\gamma}{\kappa}\sqrt{\frac{\pi C_{T,rated}}{8S^2}\left(U_r / U_H\right)^{3/2} + \left(\kappa/\delta\right)^2}\right) \\ + \ & U_H\left(1 + \frac{\gamma}{\delta}\right)^{-1}\times\left(\frac{3}{4}\frac{\gamma}{\kappa}\frac{\pi C_{T,rated}}{8S^2}\left(U_r / U_H\right)^{1/2}\left(\frac{U_r}{U_H^2}\right)\times\left(\frac{\pi C_{T,rated}}{8S^2}\left(U_r / U_H\right)^{3/2} + \left(\kappa/\delta\right)^2\right)^{-1/2}\right) \end{aligned} \ . \ (A.3)$$


With $\gamma$, $\kappa$, $\delta$, $U_r$ and $U_H$ being positive, $dU_{H,0}/dU_H$ is positive, and thereby $dU_H/dU_{H,0}$ is positive for any (positive)
value of $U_{H,0}$ which in turn means that $U_H(U_{H,0})$ is strictly monotonic. As seen, this qualitative result has been obtained
without knowing the explicit form of the function $U_H(U_{H,0})$.