# Peer review of "Towards the North Sea wind power revolution"

_Wind Energy Science, 2018_

## Referee Comment (RC1) · Anonymous Referee #1 · 3 Oct 2018

Review wes-2018-53

This paper aims to determine the optimal price and spacing for wind turbines deployed in the North Sea to provide 50% of Europe's electricity.

Assumptions made in this analysis such as Ct=0.8 and the mean wind speed of 9.7 m/s at 100 m height are unnecessary. WRF modelling would be able to provide a complete wind resource estimate for every location in the North Sea or wind speeds from satellite maps that are derived at DTU could have been applied. See e.g. Hasager, C. B., Astrup, P., Zhu, R., Chang, R., Badger, M., & Hahmann, A. N. (2016). Quarter-Century Offshore Winds from SSM/I and WRF in the North Sea and South China Sea. Remote Sensing, 8(10), [769]. DOI: 10.3390/rs8090769. Equally information about the turbines deployed is available.

The fundamental assumption is (p3): 'As scale parameters we employ $\lambda$ = 11 m/s and k = 2.2, corresponding to an average wind speed of 9.7 m/s, at a 100 m altitude. The numbers are taken as averaged values from measurements and simulations of selected locations in the North Sea (see Pena and Hahmann, 2017).' In other words, the wind resource used in the analysis doesn't vary according to the location in the North Sea, which we already know from DTU Wind Atlas (and many others) is incorrect. The mean wind speed at a given height over the North Sea in the DTU wind Atlas varies by at least 2m/s from north to south, even disregarding the impact of (land) topography, distance to coast and so on. In other words, we already know there is a $\pm$20% variation on the mean wind speed, and more on the potential power which is not utilized in this study. How big is the uncertainty introduced here? Is it greater than the uncertainty in the spacing proposed? How does this propagate through the pricing?

After further consideration in the manuscript, an expression is derived for the power density per unit area that depends on the spacing of the turbines given the rated power and an assumed power coefficient. This calculation is one that has been reworked by quite a few authors already – it might be good to cite some of them to give an idea of the range.

The section on bathymetry is quite confusing. Why can't a contour/image map or similar be provided rather than the bathymetry along a line? There are quite a few (web) services that offer (free) download of bathymetric data and it is already stated that the authors have access to this dataset.

Similarly, the cost models seem to be based on rather straightforward assumptions, some which are a bit out-of-date.

The results for power density are quite similar to a few other studies that indicate what is generally known – there is a very large uncertainty – we know the power density for wind energy is somewhere around 2-6 Wm-2 (see also Jacobsen PNAS and then a whole raft of papers by others who argue it is around 1 Wm-2) and agrees with studies
like the ons cited (by Frandsen 2009).

The capacity factors seem low here (Figure 6) – cf Wiser who states the actual average in the US (not offshore) is around 41%?

Given the assumptions in the above with a non-varying wind speed, Figure 7 is a scaled version of the bathymetry distribution in Figure 3 and Figure 4 likewise to Figure 8 (The authors state this in the text). Some of the figures are redundant or could be combined to provide a more interesting analysis.

Fundamentally then, it is incorrect then to state that Figures 9, 10 and 11 are: Figure 9: Area required to produce Europe's power demand as function of spacing and rotor diameter. Figure 10: Installed power required to produce Europe's power demand as function of spacing and rotor diameter Figure 11: Number of turbines required to produce Europe's power demand as function of spacing and rotor diameter.

because all of these depend on the assumed wind resource and will be different for different wind climates. (At the very least this should be stated in the captions).

Much of this analysis could be done by simply looking up that in 2016 Europe's electricity demand was 3.1 million GWh or 3100 TWh equivalent to (as stated in the paper) around 0.4TW and assuming wind energy extraction of 1-4 Wm-2

(https://ec.europa.eu/eurostat/statistics-explained/index.php/Electricity_production,_consumption_and_market_overview)

gives an area as stated in the manuscript of around 400,000 km-2 for wind energy extraction of 1 Wm-2 or maybe half or a third of that if the energy extraction rate is 2 Wm-2 or 3 Wm-3 respectively which you can then compare to the area of the North Sea (590,000 km2). You could assume a capacity factor of 0.4 and simply calculate the number of 13 MW turbines based on that. The number of turbines/power per unit area is given essentially as the inverse of the square of the spacing (from Frandsen).

We know already that a major factor in the cost is the actual resource – so if this is not factored in it ought to be included in the uncertainty. The scaling applied here is

about what power coefficients are assumed since wind speed variability and its impact on wake losses are not considered.

The O&M numbers in Figure 16 go against results indicated in terms of turbine lifetimes and O&M on land (do costs decrease over time?). There are a number of simplifying assumptions used such as such scaled turbine costs from 2003 and 'to arrive at reasonable realistic LCoE estimates we will, in line with Mahulja (2015), assume that cost of WT's, internal WF grid and foundations accounts for 75% of the total investment costs, which is based on experiences from the Danish Horns Rev and Nysted offshore wind farms. The remaining 25% is mainly due to electrical infrastructures, such as onshore cables and substations.' This is of course directly reflected in the results but a more sophisticated analysis would show some variation with distance to the coast etc.

If the authors decide to proceed, at the very least they must make clear that their results are entirely based on 'an average wind speed of 9.7 m/s, at a 100 m altitude' over the whole area and list the other simplifying assumptions in the other parts of the manuscript in the conclusions and the abstract and comment on how these assumptions are propagated as uncertainty. If on the other hand they are able to use the results they have available in house on the wind resource etc, this would make the work significantly more relevant.

---

## Referee Comment (RC2) · Anonymous Referee #2 · 22 Oct 2018

The main concerns about this paper are the following:

* The first concern is about a technical question. The authors propose some analytical formulas to be used in order to extrapolate some of the costs involved that seem to be plausible and extracted from papers. However, they may lack of strong experimental support in real wind farms and therefore subject to criticism.

* The second one is about the whole approach to consider the usefulness of the paper. Does it make sense to install of the turbines in the North Sea to provide energy to Europe from a technical, operational, social, environmental, and political point of view? And the answer is no. That makes the paper an interesting but theoretical one.

* A third concern is about doing a sensitivity analysis to detect which are the relevant

parameters in the impact they have on the overall computation in order to show them and their impact.

---

## Author Comment (AC1) · 7 Nov 2018

**Answers to Reviewer 1**

**Reviewer1:** This paper aims to determine the optimal price and spacing for wind turbines deployed in the North Sea to provide 50% of Europe's electricity.
**Answer:** No, as written in the abstract, it is 100% we are aiming for.

**Reviewer1:** Assumptions made in this analysis such as Ct=0.8 and the mean wind speed of 9.7 m/s at 100 m height are unnecessary. WRF modelling would be able to provide a complete wind resource estimate for every location in the North Sea or wind speeds from satellite maps that are derived at DTU could have been applied. See e.g. Hasager, C. B., Astrup, P., Zhu, R., Chang, R., Badger, M., & Hahmann, A. N. (2016). Quarter-Century Offshore Winds from SSM/I and WRF in the North Sea and South China Sea. Remote Sensing, 8(10), [769]. DOI: 10.3390/rs8090769. Equally information about the turbines deployed is available.
**Answer:** We don't agree on this statement. First, as is discussed in detail in the paper, the atmospheric model includes the influence of the wakes on the actual wind speeds encountered by the wind turbines. The assumption of CT=0.8, corresponding to a typical optimum condition for a modern wind turbine below rated speed, is absolutely necessary to include in the atmospheric wake model, see eq. (1). For turbines operating above the rated wind speed CT is taken as a variable following the expression in eq. (21). Using e.g. satellite measurements or WRF modelling would not include wake effects, and are *per se* not sufficient for predictions of the power production. We agree that including a full velocity map of the Norths Sea would be ideal. However, this requires the full area of the North Sea to be discretized in up to the order of 100.000 grid points, computing the power production of each wind turbine separately. This may be feasible, but requires a tremendous amount of work, and is clearly outside the scope of the present investigation. This could later be the subject for a more detailed research project.

**Reviewer1:** The fundamental assumption is (p3): 'As scale parameters we employ _ = 11 m/s and k = 2.2, corresponding to an average wind speed of 9.7 m/s, at a 100 m altitude. The numbers are taken as averaged values from measurements and simulations of selected locations in the North Sea (see Pena and Hahmann, 2017).' In other words, the wind resource used in the analysis doesn't vary according to the location in the North Sea, which we already know from DTU Wind Atlas (and many others) is incorrect. The mean wind speed at a given height over the North Sea in the DTU wind Atlas varies by at least 2m/s from north to south, even disregarding the impact of (land) topography, distance to coast and so on. In other words, we already know there is a 20% variation on the mean wind speed, and more on the potential power which is not utilized in this study. How big is the uncertainty introduced here? Is it greater than the uncertainty in the spacing proposed? How does this propagate through the pricing?
**Answer:** The aim of the present work has been to develop a simple model that includes all important parameters for an economic assessment of the overall figures of a massive exploitation of wind power in the North Sea. Therefore, the local variations are of less importance, and since we ideally are talking about 100.000 13 MW WT's distributed over an area of about 190.000 km$^2$, the differences will clearly be outbalanced in the computed average power production. Therefore, the uncertainty on the power production and the pricing is negligible.

**Reviewer1:** After further consideration in the manuscript, an expression is derived for the power density per unit area that depends on the spacing of the turbines given the rated power and an assumed power

coefficient. This calculation is one that has been reworked by quite a few authors already – it might be good to cite some of them to give an idea of the range.

**Answer:** Numbers on power density per unit area are given in section 3.1 (page 13), where we refer to measurements of the Danish Horns Rev and Nysted (Rødsand) wind farms, and also to theoretical values from a study by Frandsen. In these studies the values are found to vary in the range from 1.9 W/m$^2$ to 4 W/m$^2$, which are similar to the figures in the present study.

**Reviewer1:** The section on bathymetry is quite confusing. Why can't a contour/image map or similar be provided rather than the bathymetry along a line? There are quite a few (web) services that offer (free) download of bathymetric data and it is already stated that the authors have access to this dataset.

**Answer:** It would indeed be possible to use contour data. However, such a plot would, in our opinion, be less informative  and only indirectly provide the needed information for the 'optimization approach' taken in this paper. The aim of Fig. 2. is to illustrate typical values in the Southern part of the North Sea that illustrates that a large part contains water depths less than 50 m. Figs. 3 and 4 is a simple way of illustrating how much area contains a specific water depth. We show explicitly these figures, as this is exactly the data that has been used in our analysis. This is required if somebody wants to reproduce our results. In the revised version of the paper, we have made it clearer that this is actually the 'raw' data used in our analysis.

**Reviewer1:** Similarly, the cost models seem to be based on rather straightforward assumptions, some which are a bit out-of-date.

**Answer:** The various elements of the cost model are based on the most recent literature that we have access to. There are indeed a lot of information in the pertinent literature regarding cost of energy produced by wind power, but most of it is not detailed enough to be used in an actual cost model. It is clear that CoE is always decreasing due to more efficient production methods, and the numbers we use are somewhat conservative. Nevertheless, they correspond surprisingly well to existing CoE prices for existing offshore wind farms, which is also noted in the paper.

**Reviewer1:** The results for power density are quite similar to a few other studies that indicate what is generally known – there is a very large uncertainty – we know the power density for wind energy is somewhere around 2-6 Wm-2 (see also Jacobsen PNAS and then a whole raft of papers by others who argue it is around 1 Wm-2) and agrees with studies like the ones cited (by Frandsen 2009).

**Answer:** Power density is always dependent on the specific site, the size of the turbines, and the distance between them. Our model predicts a maximum of about 7-8 W/m$^2$ for very densely located turbines (1.5-2 diameters). The paper by Jacobsen (PNAS) did not include wake effects and other more pessimistic papers as the one by Adams and Keith (ERL, 2013) are for the Great Plains, which not really is comparable to the North Sea. The values of 1-2 W/m$^2$ for turbines of mutual distances of about 7 diameters located in the North Sea correspond well to the numbers by Frandsen (this is actually mentioned in the paper).

**Reviewer1:** The capacity factors seem low here (Figure 6) – cf Wiser who states the actual average in the US (not offshore) is around 41%?

**Answer**: We do not agree that our capacity factors are low. In fact, they corresponds to what is normally seen in practice. The low numbers appearing in the left part of Fig. 6. is due to wake effects for

unrealistically low spacing of the turbines. According to the report by Wiser (Ryan Wiser & Mark Bolinger: '2016 Wind Technologies Market Report: Summary', Lawrence Berkeley National Laboratory, August 2017, page 34) the average capacity factor in the US is about 33%, which corresponds well to our numbers for a spacing of 7 diameters in a wind farm consisting of 3.2 MW wind turbines.

**Reviewer1:** Given the assumptions in the above with a non-varying wind speed, Figure 7 is a scaled version of the bathymetry distribution in Figure 3 and Figure 4 likewise to Figure 8 (The authors state this in the text). Some of the figures are redundant or could be combined to provide a more interesting analysis.
**Answer:** For the ease of the reader to understand the figures, we prefer to maintain Figs. 8 and 9, since they have different ordinate values and give production data, whereas Figs. 3 and 4 are bathymetric data.

**Reviewer1:** Fundamentally then, it is incorrect then to state that Figures 9, 10 and 11 are: Figure 9: Area required to produce Europe's power demand as function of spacing and rotor diameter. Figure 10: Installed power required to produce Europe's power demand as function of spacing and rotor diameter Figure 11: Number of turbines required to produce Europe's power demand as function of spacing and rotor diameter. Because all of these depend on the assumed wind resource and will be different for different wind climates. (At the very least this should be stated in the captions).
**Answer:** We agree that it should be mentioned that the numbers depend on the wind climate and thereby (naturally) are specific for the North Sea. We have accordingly included this assumptions in the figure caption.

**Reviewer1:** Much of this analysis could be done by simply looking up that in 2016 Europe's electricity demand was 3.1 million GWh or 3100 TWh equivalent to (as stated in the paper) around 0.4TW and assuming wind energy extraction of 1-4 Wm-2 (https://ec.europa.eu/eurostat/statistics-explained/index.php/Electricity_production,_consumption_and_market_overview) gives an area as stated in the manuscript of around 400,000 km-2 for wind energy extraction of 1 Wm-2 or maybe half or a third of that if the energy extraction rate is 2 Wm-2 or 3 Wm-3, respectively, which you can then compare to the area of the North Sea (590,000 km2). You could assume a capacity factor of 0.4 and simply calculate the number of 13 MW turbines based on that. The number of turbines/power per unit area is given essentially as the inverse of the square of the spacing (from Frandsen).
**Answer:** We agree partly on this. However, the analysis we provide is much more detailed than this, as we take into account the influence of spacing, and the size and operation of the turbines on the power density. Furthermore, the analysis is combined with a cost model, which highlights the fact that it is possible to increase the power density by erecting the turbines more closely (as e.g. for the Lillgrund wind farm), but on the expenses that the CoE increases. So the analysis highlights the compromise between CoE and the need for a specific production on a given sea area. This is illustrated in Fig. 14, where it is shown that there is a distinct minimum, where the lowest CoE is reached for S=8.

**Reviewer1:** We know already that a major factor in the cost is the actual resource – so if this is not factored in it ought to be included in the uncertainty. The scaling applied here is about what power coefficients are assumed since wind speed variability and its impact on wake losses are not considered.
**Answer:** It is not correct when the reviewer states that wind speed variability and its impact on wake losses are not considered. In fact, the analysis described in pp. 6-9 is about how to include the variability

into wake losses. To our knowledge, this is first time that the wind statistics is modeled directly into the wake model.

**Reviewer1:** The O&M numbers in Figure 16 go against results indicated in terms of turbine lifetimes and O&M on land (do costs decrease over time?).
**Answer**: We are not sure which conflict this statement refers to. The O&M costs used in the paper don't have a memory, but relates to the loading of the wind farm turbines. More specifically, the wind turbine loading is assumed to relate directly to the wake loss (i.e. significant wake losses imply significantly increased wind turbine loads).

**Reviewer1:** There are a number of simplifying assumptions used such as such scaled turbine costs from 2003 and 'to arrive at reasonable realistic LCoE estimates we will, in line with Mahulja (2015), assume that cost of WT's, internal WF grid and foundations accounts for 75% of the total investment costs, which is based on experiences from the Danish Horns Rev and Nysted offshore wind farms. The remaining 25% is mainly due to electrical infrastructures, such as onshore cables and substations.' This is of course directly reflected in the results but a more sophisticated analysis would show some variation with distance to the coast etc.
**Answer:** We agree, that a more sophisticated analysis, including variations with distance to the cost etc, would give a more detailed local analysis of the costs. However, the aim of the paper is to give gross figures associated with a massive exploitation of the North Sea, and not detailed and a much more computationally consuming analyses. For an isolated wind farm, it is correct, that the cost of onshore cables depends on distance to shore. However, for the massive exploitation of North Sea wind energy considered in the paper, it would, in the authors opinion, be misleading to consider each 'sub-cluster' of wind turbines as isolated. We believe, that synergy effects can be harvested by 'smart' collective onshore cable logistics interconnecting a large number of wind turbine 'sub-clusters', and consequently that the dependence of onshore cabling on distance to shore can be alleviated compared to isolated wind farms.

**Reviewer1:** If the authors decide to proceed, at the very least they must make clear that their results are entirely based on 'an average wind speed of 9.7 m/s, at a 100 m altitude' over the whole area and list the other simplifying assumptions in the other parts of the manuscript in the conclusions and the abstract and comment on how these assumptions are propagated as uncertainty. If on the other hand they are able to use the results they have available in house on the wind resource etc, this would make the work significantly more relevant.
**Answer**: We will follow the recommendation of the reviewer and make the simplifying assumptions and their consequences more clear in the final paper. However, regarding the wind resource it should be mentioned that the applied average estimate is based on detailed North Sea WRF computations, with results extracted for locations on a (North to South) line approximately 100km West of the Jutland at 100m altitude (http://orbit.dtu.dk/en/publications/30year-mesoscale-model-simulations-for-the-noise-from-wind-turbines-and-risk-of-cardiovascular-disease-project(e0fb7fc8-8c5b-4939-8b59-7da339dbc710).html). These data refer a 30-year time horizon, and we believe that the involved spatial and temporal averaging assures estimates with very little uncertainty.

**Answers to Reviewer 2**

**Reviewer2:** The first concern is about a technical question. The authors propose some analytical formulas to be used in order to extrapolate some of the costs involved that seem to be plausible and extracted from papers. However, they may lack of strong experimental support in real wind farms and therefore subject to criticism.

**Answer:** We agree that it indeed is difficult to validate properly the cost model as very little data is available. However, we found some information regarding the Danish Rødsand wind farm, where the computed cost price is quite similar to the actual one (8 €cents/kWh). This is discussed in the paper (page 15). If the reviewer is aware of other data, we would of course be very interested in getting the numbers.

**Reviewer2:** The second one is about the whole approach to consider the usefulness of the paper. Does it make sense to install of the turbines in the North Sea to provide energy to Europe from a technical, operational, social, environmental, and political point of view? And the answer is no. That makes the paper an interesting but theoretical one.

**Answer:** There is actually an increasing interest in exploiting more the potential of the wind power in the North Sea. This can be seen from the increasing interests of developers operating in the North Sea and from international political memoranda (see e.g. https://northsearegion.eu/northsee/e-energy/transnational-energy-cooperation-between-north-sea-countries/ or https://windeurope.org/wp-content/uploads/files/policy/topics/offshore/Offshore-Wind-Statement-of-Intent-signed.pdf). Furthermore, we completely agree that operational, social, environmental, and political aspects are also very important. There has been some introductory investigations on these aspects (see e.g. http://www.eera-dtoc.eu/). The aim of the present contribution is merely a statement that it technically and economically is feasible. The next step will then be to include socio-economic, operational, environmental, and political aspects.

**Reviewer2:** A third concern is about doing a sensitivity analysis to detect which are the relevant parameters in the impact they have on the overall computation in order to show them and their impact.

**Answer:** We believe that the wind resource estimate is very robust (c.f. last answer to reviewer 1) with very little statistical uncertainty, and consequently a sensitivity study based on the wind resource would not make much sense in the authors opinion. Another aspect is the wind farm layout, where a simple layout pattern is assumed. An optimized wind farm layout from an economic perspective is doable, but, for the massive exploitation of wind described in the paper, it is a tremendous computational task, involving detailed unsteady flow field simulations across the entire North Sea coupled with aeroelastic simulations of each turbine. This is clearly outside the scope of the present paper. A third aspect is the turbine price. In the study we have used 2003 prices of wind turbines corrected for inflation. Here, it may be argued, that technological breakthroughs could lower the price over time, thus making a sensitivity study of the results with respect to turbine prices meaningful.